# Giant anisotropic photonics in the 1D van der Waals semiconductor fibrous red phosphorus

Luojun Du [1,10 ✉], Yanchong Zhao [2,3,10], Linlu Wu[4,10], Xuerong Hu [1,5,10], Lide Yao[6], Yadong Wang [1], Xueyin Bai [1], Yunyun Dai[1], Jingsi Qiao [4,7], Md Gius Uddin[1], Xiaomei Li[2,3], Jouko Lahtinen [6], Xuedong Bai [2,3,8], Guangyu Zhang [2,3,8], Wei Ji [4 ✉] & Zhipei Sun [1,9 ✉]

A confined electronic system can host a wide variety of fascinating electronic, magnetic, valleytronic and photonic phenomena due to its reduced symmetry and quantum confinement effect. For the recently emerging one-dimensional van der Waals (1D vdW) materials with electrons confined in 1D sub-units, an enormous variety of intriguing physical properties and functionalities can be expected. Here, we demonstrate the coexistence of giant linear/nonlinear optical anisotropy and high emission yield in fibrous red phosphorus (FRP), an exotic 1D vdW semiconductor with quasi-flat bands and a sizeable bandgap in the visible spectral range. The degree of photoluminescence (third-order nonlinear) anisotropy can reach 90% (86%), comparable to the best performance achieved so far. Meanwhile, the photoluminescence (third-harmonic generation) intensity in 1D vdW FRP is strong, with quantum efficiency (third-order susceptibility) four (three) times larger than that in the most well-known 2D vdW materials (e.g., $MoS_2$). The concurrent realization of large linear/nonlinear optical anisotropy and emission intensity in 1D vdW FRP paves the way towards transforming the landscape of technological innovations in photonics and optoelectronics.

---

[1] Department of Electronics and Nanoengineering, Aalto University, Tietotie 3, Finland. [2] Beijing National Laboratory for Condensed Matter Physics; Key Laboratory for Nanoscale Physics and Devices, Institute of Physics, Chinese Academy of Sciences, Beijing, China. [3] School of Physical Sciences, University of Chinese Academy of Sciences, Beijing, China. [4] Beijing Key Laboratory of Optoelectronic Functional Materials & Micro-Nano Devices, Department of Physics, Renmin University of China, Beijing, P.R. China. [5] Institute of Photonics and Photon Technology, Northwest University, Xi'an, China. [6] Department of Applied Physics, Aalto University, Aalto, Finland. [7] Centre for Advanced 2D Materials and Graphene Research Centre, National University of Singapore, Singapore, Singapore. [8] Songshan Lake Materials Laboratory, Dongguan, China. [9] QTF Centre of Excellence, Department of Applied Physics, Aalto University, Aalto, Finland. [10] These authors contributed equally: Luojun Du, Yanchong Zhao, Linlu Wu, Xuerong Hu. ✉email: luojun.du@aalto.fi; wji@ruc.edu.cn; zhipei.sun@aalto.fi

Confined electronic materials with reduced symmetry and quantum confinement effect offer a fascinating platform to realize a wide variety of exotic electrical, optical, and magnetic properties, and are revolutionizing the basic scientific research, technological innovation, and our daily life. One of the remarkable examples is two-dimensional (2D) van der Waals (vdW) crystals with electrons confined in 2D atomic sheets[1,2]. A rich variety of fascinating physical phenomena and quantum-phase transitions have thus far been demonstrated in 2D vdW materials, including but not limited to strong in-plane anisotropy[3–8], nonlinear optics[9–11], quantum-anomalous Hall effect[12,13], chiral quasiparticles[14–16], exciton Bose–Einstein condensation[17,18], and unconventional superconductivity[19,20], initiating an exciting 2D era of condensed-matter physics, material science, and enormous applications[21–23]. Apart from 2D vdW materials, recent advances have demonstrated another type of confined electronic material: 1D vdW crystal[24–28]. More significantly, the electrons in 1D vdW crystals are further confined in 1D subunits (e.g., tubes, chains, and ribbons), which are bonded with each other via the vdW interactions[24]. As a result, a broad range of emergent physical phenomena and functionalities can be expected in 1D vdW materials. For instance, the distinctly different bonding features between the intrachain (e.g., strong covalent bonding) and interchain directions (e.g., weak vdW interactions) in 1D vdW materials strongly break the threefold rotational symmetry and hence would naturally enable the giant anisotropy[29–31]. Indeed, a record level of anisotropic transport properties has been recently reported in 1D vdW crystal $Sb_2Se_3$[27]. However, in marked contrast to the well-studied 2D vdW materials, up to now, only a few 1D vdW materials (e.g., elemental Te[24,25], $Sb_2Se_3$[27], and $MoS_2$–BN–carbon coaxial nanotubes[32–34]) have been experimentally uncovered, mainly focusing on the electrical transport properties. Exploring new 1D vdW materials and systematically studying their properties beyond the transport properties would be of great importance for both basic research and fascinating technological applications in electronics, photonics and optoelectronics.

Here, we report the giant anisotropic optical properties and large emission intensities in a fascinating 1D vdW material: fibrous red phosphorus (FRP), an exotic allotropic modification of elemental phosphorus. Through angle- and polarization-resolved measurements of photoluminescence (PL), Raman, and third harmonic generation (THG), we uncover that 1D vdW FRP exhibits an unusually high anisotropy in its PL emission, phononic and third-order nonlinear optical responses. Simultaneously, we demonstrate that 1D vdW FRP is a semiconductor with quasi-flat bands and a sizeable bandgap in the visible spectral range, enabling strong PL (THG) emission intensity with an enhancement factor of 40 (100) as compared with these in the well-known molybdenum disulfide ($MoS_2$). The coexistence of an unusually large linear/nonlinear anisotropy and strong optical responses in 1D vdW FRP presents the possibilities for a wide variety of advanced functionalities in photonics and optoelectronics, such as phase-matching elements, communications, polarizers, sensing, and polarization-sensitive photodetectors.

## Results

**Structure characterization.** Elemental phosphorus displays several different allotropic modifications[35], such as black phosphorus (BP), which has been extensively studied and ushered in the fascinating era of in-plane anisotropy in 2D vdW materials[4,8,30]. Remarkably, recent studies uncover another phosphorus allotrope, namely FRP or red phosphorus IV, which shows excellent physical and chemical properties, e.g., relatively high carrier mobility of 300 $cm^2V^{-1}s^{-1}$ and large hydrogen-evolution efficiency[35–38]. FRP is typically composed of bundles of parallel double pentagonal-shaped phosphorus tubes along the $b$ direction (Fig. 1a–c)[35], different from the puckered atomic layers in 2D vdW black phosphorus. While the averaged separation among the double-tube pairs is ~6.77 Å, the two pentagonal-shaped phosphorus tubes are covalently bonded by a phosphorus dimer with a bond length of ~2.27 Å, offering an intertube spacing of ~5.60 Å. The pentagonal-tube pairs are bound with each other in a and $c$ directions by weak vdW interactions, which form two surfaces in the $ab$ and $bc$ planes upon cleavage with surface energies of ~21 and 26 meV/Å$^2$, respectively. Obviously, FRP belongs to the exotic 1D vdW crystal with a completely different bonding nature between distinct directions, i.e., covalent interaction versus vdW force. Consequently, the threefold rotational symmetry is explicitly broken in 1D vdW FRP, enabling strong anisotropic physical properties in principle.

Bulk FRP crystals are synthesized through flux zone growth technology (2D semiconductors, see "Methods" for details). To confirm the chemical composition, crystal structure and quality, the synthesized FRP crystals are characterized by powder X-ray diffraction (XRD), X-ray photoelectron spectroscopy (XPS), energy-dispersive X-ray spectroscopy (EDS) and high-resolution transmission electron microscopy (HRTEM). The upper panel of Fig. 1d shows the experimental results of XRD, which can be well described by the theoretical simulations (the lower panel of Fig. 1d) using the lattice parameters of 1D vdW FRP in ref. 35 ($a$ = 12.198 Å, $b$ = 12.986 Å, and $c$ = 7.075 Å; $\alpha$ = 116.99°, $\beta$ = 106.31°, and $\gamma$ = 97.91°). This indicates that our samples are indeed the 1D vdW FRP. XPS measurements (Supplementary Fig. 1) show two peaks with binding energies of ~130.6 and 131.2 eV, which correspond to $2p_{3/2}$ and $2p_{1/2}$ of phosphorus–phosphorus bonds in 1D vdW FRP, respectively[5,39]. The EDS spectrum (Supplementary Fig. 2) shows that except for the signals from the phosphorus element and TEM copper grid, no other signals can be observed, evidencing that the 1D vdW FRP crystals are of high purity. Figure 1e presents the EDS mapping, demonstrating the homogeneous distribution of phosphorus element. The selected-area electron-diffraction (SAED) pattern with clear diffraction spots (Supplementary Fig. 3) reveals that the 1D vdW FRP is highly crystalline. Figure 1f exhibits the typical HRTEM image with sharp lattice fringes, further confirming the single-crystalline nature. The corresponding fast Fourier transformation (FFT) image is shown in the inset of Fig. 1f. The crystal planes correlating with the brightest spots are labeled, i.e., ($20\bar{1}$), ($04\bar{1}$) and ($24\bar{2}$). The corresponding measured $d$-spacing values are $d_{(20\bar{1})}$ = 5.13 Å, $d_{(04\bar{1})}$ = 3.23 Å and $d_{(24\bar{2})}$ = 2.63 Å, which are in good agreement with the calculated values (5.14 Å, 3.21 Å and 2.62 Å, respectively) based on the lattice constant in ref. 35. The inter-planar angles between ($20\bar{1}$) and ($04\bar{1}$)/($24\bar{2}$) are ~85°/54.3°. Based on the indexed reflection spots in the FFT pattern, it can be concluded that the HRTEM image in Fig. 1f is taken along the [214] zone axis approximately perpendicular to the $b$ direction of 1D vdW FRP crystal (as marked by the arrow in Fig. 1f). This means that 1D vdW FRP grows along the $b$ axis. Figure 1g shows the enlarged HRTEM image from the highlighted area outlined in the white box in Fig. 1f. The lattice fringes lying parallel to each other are the iconic feature of 1D vdW FRP[35]. The interplanar distance extended along $b$ direction is ~5.13 Å, corresponding to ($20\bar{1}$) plane. Note that the enlarged HRTEM image displays a perfect atomic-scale crystal structure devoid of typical defects (such as vacancies, dislocations and stacking faults), confirming the high quality of our 1D vdW FRP crystals.

**Strong PL emission.** Ultrathin FRP flakes are exfoliated onto 300 nm $SiO_2$/Si substrates from the bulk crystals using a modified

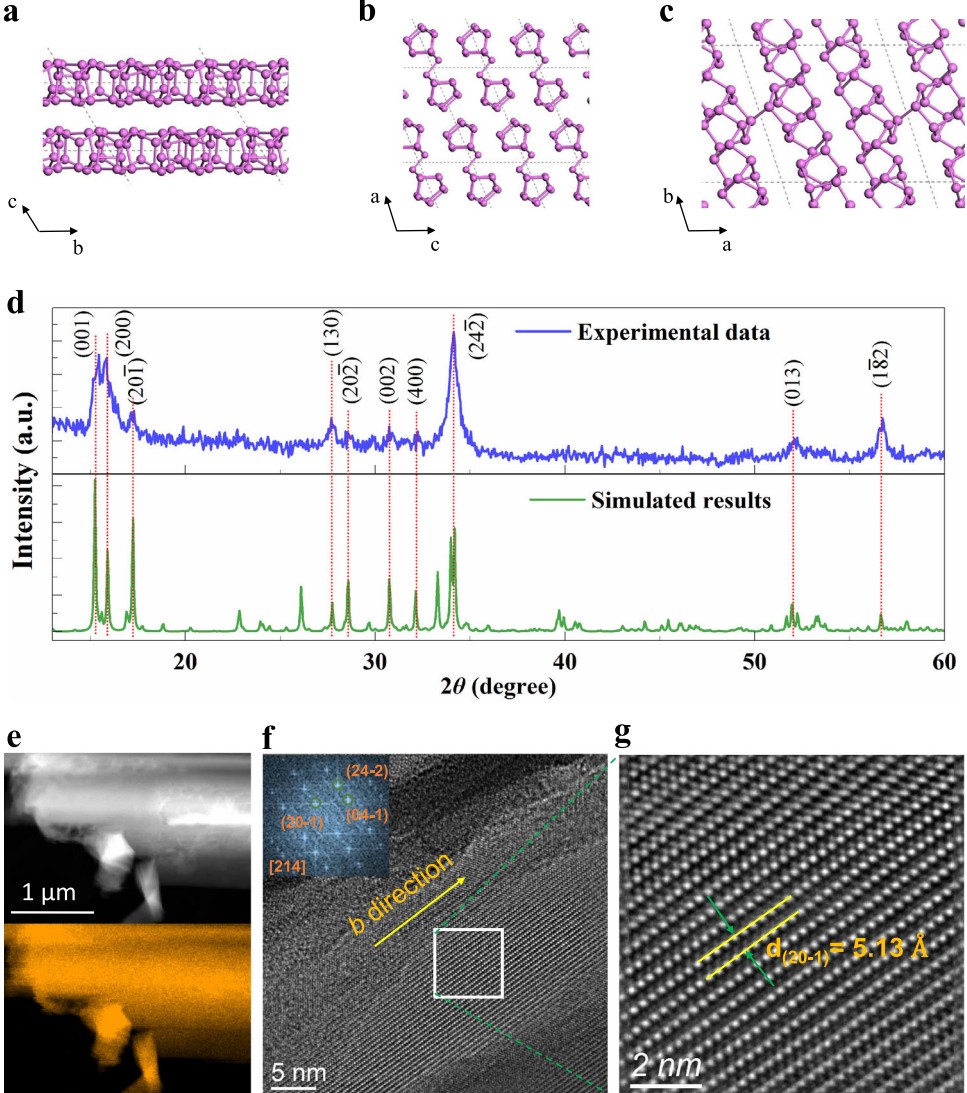

**Fig. 1 Characterization of 1D vdW FRP crystal. a–c** Schematic structural view of 1D vdW FRP crystal along the *a* (**a**), *b* (**b**) and *c* (**c**) directions. DFT-calculated lattice parameters along the *a*, *b*, and *c* directions are 12.37 Å, 13.11 Å, and 7.14 Å, respectively. The angles *α* (between *b* and *c*), *β* (between *a* and *c*), and *γ* (between *a* and *b*) are ~117.07°, 106.13°, and 97.81°, respectively. **d** The experimentally measured XRD diffractogram (upper panel) and simulated patterns (lower panel). **e** The low-magnification TEM image of 1D vdW FRP crystal (upper panel) and the corresponding EDS mapping of the phosphorus element (lower panel). **f** HRTEM image from the [214] zone axis, showing exclusively strands lying parallel to each other. The inset is the corresponding FFT pattern. **g** Enlarged HRTEM image taken from the area marked by the white outline in **f**.

mechanical exfoliation method where the substrate has been subjected to oxygen plasma to remove ambient adsorbates and enhance the adhesion force (see Methods for details)[40,41]. The exfoliated 1D vdW FRP samples are usually presented as ribbon shape with a large ratio of the length to the width. A typical white-light microscope image is shown in Supplementary Fig. 4. The upper panel of Fig. 2a shows the height image of a typical exfoliated 1D vdW FRP flake obtained from the tapping-mode atomic force microscopy (AFM) scanning. In stark contrast to the well-studied 2D vdW materials where uniform few-layer samples with large-size can be easily obtained, the exfoliated 1D vdW FRP flakes are usually not uniform and contain several areas of distinct thicknesses (the width of each area is usually less than 300 nm). This can be understood as that there are two directions (*a* axis and *c* axis) linked through the vdW force in 1D vdW FRP, leading to the similar surface energy between *ab* and *bc* planes. The height profile (lower panel of Fig. 2a) taken along the black dotted line in the upper panel of Fig. 2a indicates that the thinnest

thickness of the exfoliated 1D vdW FRP flake is ~40 nm. It is worth noting that it is difficult to obtain atomically thin FRP by mechanical exfoliation. Considering that liquid-phase exfoliation with freezing–thawing strategy has successfully produced ultra-thin $Sb_2Se_3$ (a 1D vdW material)[27], it may help to get atomically thin ribbons/fibers of FRP and deserves further studies.

Figure 2b presents the nonpolarized PL spectra at room temperature for the exfoliated 1D vdW FRP flake shown in Fig. 2a, excited by a 2.33-eV continuous-wave laser in vacuum environments. Remarkably, the PL spectra of 1D vdW FRP consist of multiple peaks. Through Lorentzian fitting, we can clearly distinguish eight sharp PL peaks, labeled as P1–P8. Intriguingly, the optical bandgap, e.g., from ~1.46 eV (P1) to 2.05 eV (P8), is in the fascinating visible and near-infrared spectral range, satisfying numerous existing and emerging technological innovations in photonics and optoelectronics[42]. Of particular importance, the exfoliated 1D vdW FRP flakes possess strong PL emission intensity. The integral PL intensity in the exfoliated 1D

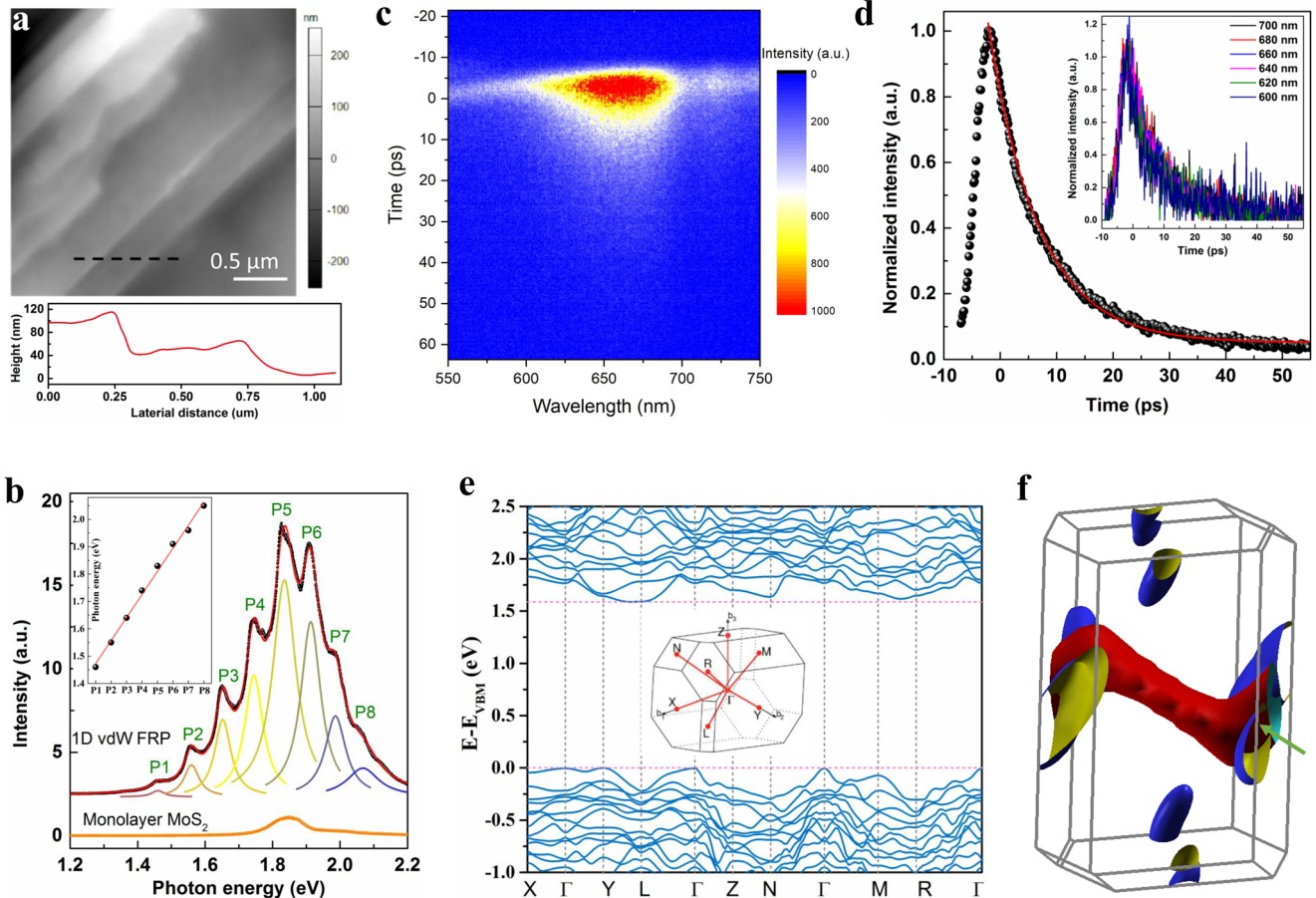

**Fig. 2 Ultrastrong PL emission in 1D vdW FRP. a** Upper panel: the typical AFM image of an exfoliated 1D vdW FRP flake, exhibiting several different thicknesses. Lower panel: height profiles taken along the dotted black line in the upper panel. **b** Nonpolarized PL spectra (black) for the exfoliated 1D vdW FRP flake in **a**, excited by a 532 nm (2.33-eV) excitation laser at room temperature. The PL spectra of 1D vdW FRP flake are fit to a sum of Lorentzians, from which we can obtain eight sharp PL peaks, marked as P1–P8. The inset is the energies for the eight PL peaks. Orange line is the PL spectra of monolayer MoS$_2$ measured at the same condition. Traces are vertically offset for clarity. **c** Time-resolved PL spectra measured with a streak camera. **d** The time evolution of normalized integral intensity from 600 nm to 700 nm. Red line is a single-exponential fit. The inset is the time evolution of normalized emission intensity at different wavelengths. **e** Theoretically calculated band structure of 1D vdW FRP. The inset shows the BZ path of primitive cells. **f** The mapping of Fermi surface for 1D vdW FRP. Red and cyan isosurfaces represent the highest valence band, and blue and yellow isosurfaces denote the lowest conduction band. The expansion of energy is 40 meV.

vdW FRP flake, including all eight peaks, shows an enhancement factor of more than 40, compared with that in the mechanically exfoliated monolayer MoS$_2$, which is one of the most well-known direct-gap 2D semiconductors with large PL quantum efficiency[43]. Considering that monolayer MoS$_2$ possesses an absorbance of ~10% under 2.33-eV excitation[44], the PL quantum efficiency of FRP should be more than four times larger than that of monolayer MoS$_2$, even if we assume that the absorption of FRP is 100%. Figure 2c shows the time-resolved PL spectra measured with a streak camera (see Methods for details). The inset of Fig. 2d is the normalized emission intensity as a function of time at different wavelengths. We can see clearly that the time evolution of normalized emission intensity at different wavelengths coincides with each other, indicating the same lifetime for PL emissions at different wavelengths. Figure 2d presents the time evolution of normalized integral intensity from 600 nm to 700 nm. Through fitting with a single-exponential function, we extract that the PL lifetime in FRP is about 10 ps. Note that the PL lifetime in FRP (~10 ps) is much longer than the lifetime of direct exciton in WSe$_2$ (~150 fs)[45] and close to the lifetime of indirect exciton/phonon replicas in bilayer WSe$_2$ (~100 ps)[46].

To demystify the origin of multiple PL peaks and strong PL emission intensity observed in the 1D vdW FRP crystal, we perform the electronic structure calculations within the context of density-functional theory (DFT) with the optB88-vdW functional for the exchange potential (see Methods for details). Figure 2e plots the theoretical electronic band structures of 1D vdW FRP. The conduction band minimum (CBM) and valence band maximum (VBM) are located near the L and Γ/Y points of the Brillouin zone (BZ), respectively. Note that the eigenvalues at the Y and Γ points in the valence band are nearly degenerated, with an energy difference of less than 4.8 meV. This suggests that 1D vdW FRP is an indirect-bandgap semiconductor with a bandgap of ~1.57 eV. Note that the adopted optB88-vdW functional usually underestimates the bandgap of a semiconductor, primarily due to the over-estimated delocalization of electrons. The modified Becke–Johnson (mBJ) functional with such issue corrected shall predict a reliable fundamental bandgap[47]. Indeed, the mBJ functional yields an indirect fundamental bandgap of ~1.80–2.11 eV for FRP (Supplementary Fig. 8), which is very close to the energy of the P8 peak (2.05 eV, Fig. 2b). Because of the indirect bandgap nature, the PL emission requires the

participation of phonons or other elementary excitations to satisfy the conservation of momentum. One plausible origin for the multiple PL peaks with equal energy spacing (inset of Fig. 2b) is phonon replicas, considering that half of the energy separation (~ 42 meV) can match well with the phonon energy (Fig. 5a) and other elementary excitations (such as magnon, polaron, and plasmon) can be largely ruled out due to the nonmagnetic semiconducting nature of 1D vdW FRP. Further in-depth studies, however, are required to fully understand its underlying origin. Moreover, our calculations show that the bandwidth of the valence band along X–Γ–Y direction is very small, indicating the quasi-flat band with a high density of states, which can be viewed as an analogy to the van Hove singularity in 1D materials. Such weak dispersion in the valence band originates fundamentally from the 1D-confinement effect of electrons and is further confirmed by the three-dimensional mapping of Fermi surface (Fig. 2f). The ultra-high density of states may balance the suppressed transition probability due to the indirect-bandgap and be responsible for the large PL intensity. In addition, given an energy broadening of ~40 meV, the isosurfaces of conduction and valence bands partially superpose near Y point of the BZ (pointed by the green arrow in Fig. 2f). This indicates the possibility of phonons with small momentum participating in the PL process for the large response.

**Giant linear dichroism**. As mentioned above, the intrinsic threefold rotational symmetry is broken in 1D vdW FRP crystal, which would enable strong anisotropic physical properties. To confirm this, we first perform the angle- and polarization-resolved PL measurements (see Methods for details) with a 2.33-eV excitation laser to study the intriguing anisotropic PL responses since 1D vdW FRP crystal harbors strong PL emission efficiency. Figure 3a presents the polarization-resolved PL spectra of an exfoliated 1D vdW FRP sample under four different polarization configurations. Note that the $x$ ($y$) axis is defined as along (perpendicular to) the axial direction of the 1D subunit of phosphorus tube, i.e., the $b$ direction of 1D vdW FRP crystal (Supplementary Fig. 4)[48]. Strikingly, the PL emission intensities strongly depend on the polarization configuration. When both the excitation and detection polarizations are aligned with the axial (radial) direction of the 1D subunit of phosphorus tube, the brightest (darkest) PL emission occurs. This unequivocally demonstrates the anisotropy and 1D nature of PL emission[3,49].

Figure 3b–h presents the polar plots in sequence for polarization angle $\theta$-dependent emission intensities of PL peaks P2–P8 detected under copolarized (linearly parallel, red) and cross-polarized (linearly perpendicular, blue) configurations ($\theta$ is defined as the angle between the polarization of the excitation laser and $b$ direction of 1D vdW FRP crystal). For PL peaks P2–P8, the PL intensities vary periodically with the angle $\theta$ and can be well fitted by a $\cos^2\theta$ ($\sin^2\theta$) function plus an offset under colinearly (cross-linearly) polarized configuration, as shown by the solid red (blue) lines. The results further confirm the highly anisotropic PL emission. To quantify the magnitude of PL anisotropy, we define the degree of linear dichroism as $\rho = \frac{I_{co} - I_{cross}}{I_{co} + I_{cross}}$, where $I_{co}$ ($I_{cross}$) denotes the PL intensity detected copolarized (cross-polarized) to the laser polarization[8,49,50]. The inset of Fig. 3i shows the $\theta$-driven evolution of linear dichroism for the PL peak P6, which can be described well by the function of $\cos2\theta$ plus a small offset. Remarkably, the linear dichroism of PL peak P6 can reach more than 90%, being comparable to the highest values obtained among the well-known quasi-1D van der Waals materials (Fig. 3i)[3,6,50–55]. The linear dichroism of other PL peaks can be found in Supplementary Fig. 5. Apart from PL peak P1, a high degree of linear dichroism can exist in all other PL

peaks. The unusually large linear dichroism within the visible range, in combination with giant PL emission intensity, would provide a firm basis for the development of numerous advanced photonic and optoelectronic applications.

**Resonance-enhancement effect of THG**. Beyond the strong in-plane anisotropy of the PL process, the broken threefold rotational symmetry enabled by the extreme difference of bonding nature between intrachain and interchain directions would also lead to highly anisotropic nonlinear optical responses, e.g., THG and inelastic Raman scattering. For THG, it belongs to one of the most widely studied nonlinear optical processes in which new photons with energy triple that of the incident photons are generated (inset of Fig. 4a), and plays a pivotal role in numerous technological advances (e.g., lasers, frequency converters, and electro-optic modulators)[9]. According to the generally accepted nonlinear optical principles[9,56], THG intensity exhibits a cubic dependence on the excitation power and can be expressed as $I_{THG} \propto |P(\omega)|^3$, where $I_{THG}$ and $P(\omega)$ are the THG intensity and excitation power, respectively. Figure 4a is the natural logarithm plot of THG intensity as a function of the excitation power and demonstrates that the exponent is about 2.85, in good agreement with the theoretical value of 3. The upper panel of Fig. 4b presents the THG spectra of 1D vdW FRP, measured under different excitation wavelengths with the same average power of ~2.6 μW (46 GW/cm² intensity). Strikingly, the THG intensities are strongly dependent on the excitation photon energy and show three local maximum points with THG photon energies of ~2.86 eV (433 nm), 2.55 eV (486 nm), and 2.43 eV (511 nm). Such three maximum points are in good line with the absorption peaks uncovered by linear reflectance contrast spectra (the lower panel of Fig. 4b) and theoretically calculated absorption spectra (Supplementary Fig. 6), indicating the giant resonance-enhancement effect from interband transitions[9,57]. Note that the energies of these three THG resonance peaks are larger than the PL energies (Fig. 2b), revealing transitions associated with higher energy bands. Remarkably, the THG responses of the 1D vdW FRP flake under the resonance condition are strong and around two orders of magnitude larger than that generated in few-layer MoS₂ under the same condition (Fig. 4c). In order to better compare the THG between FRP and MoS₂, we derive the THG third-order susceptibility $\chi^{(3)}$

$$\chi^{(3)} = \frac{4\varepsilon_0 c^2}{3\omega d}\sqrt{n_\omega^3 n_{3\omega}\frac{I_{3\omega}}{I_\omega^3}} \tag{1}$$

where $\varepsilon_0$, $c$, and $d$ are the permittivity of vacuum, speed of light, and sample thickness, respectively[58]. $n_\omega$ ($n_{3\omega}$) and $I_\omega$ ($I_{3\omega}$) are the refractive index at frequency $\omega$ ($3\omega$) and pump (THG) intensity, respectively. Based on Eq. (1), we obtain that the THG third-order susceptibility of FRP is $\chi^{(3)}(FRP) = \sim 2.67 \times 10^{-18} m^2 V^{-2}$, three times larger than that of MoS₂ ($\chi^{(3)}(MoS_2) = \sim 8.99 \times 10^{-19} m^2 V^{-2}$). Note that calculated refractive indexes $n_\omega$ and $n_{3\omega}$ (Supplementary Fig. 7) are used to derive the THG third-order susceptibility of FRP because of the absence of experimental results. In addition, the THG intensity in 1D vdW FRP is 75 (60) times stronger than that in gold (Si), and also four times larger than that observed in the well-known third-order nonlinear commercial bulk GaAs[9,56].

**Anisotropic THG**. Further, we perform angle- and polarization-resolved THG measurements to uncover the fascinating anisotropic third-order nonlinear optical response in 1D vdW FRP (see Methods for details). Figure 4d shows the polar plot for polarization angle $\theta$-dependent THG intensities under copolarized

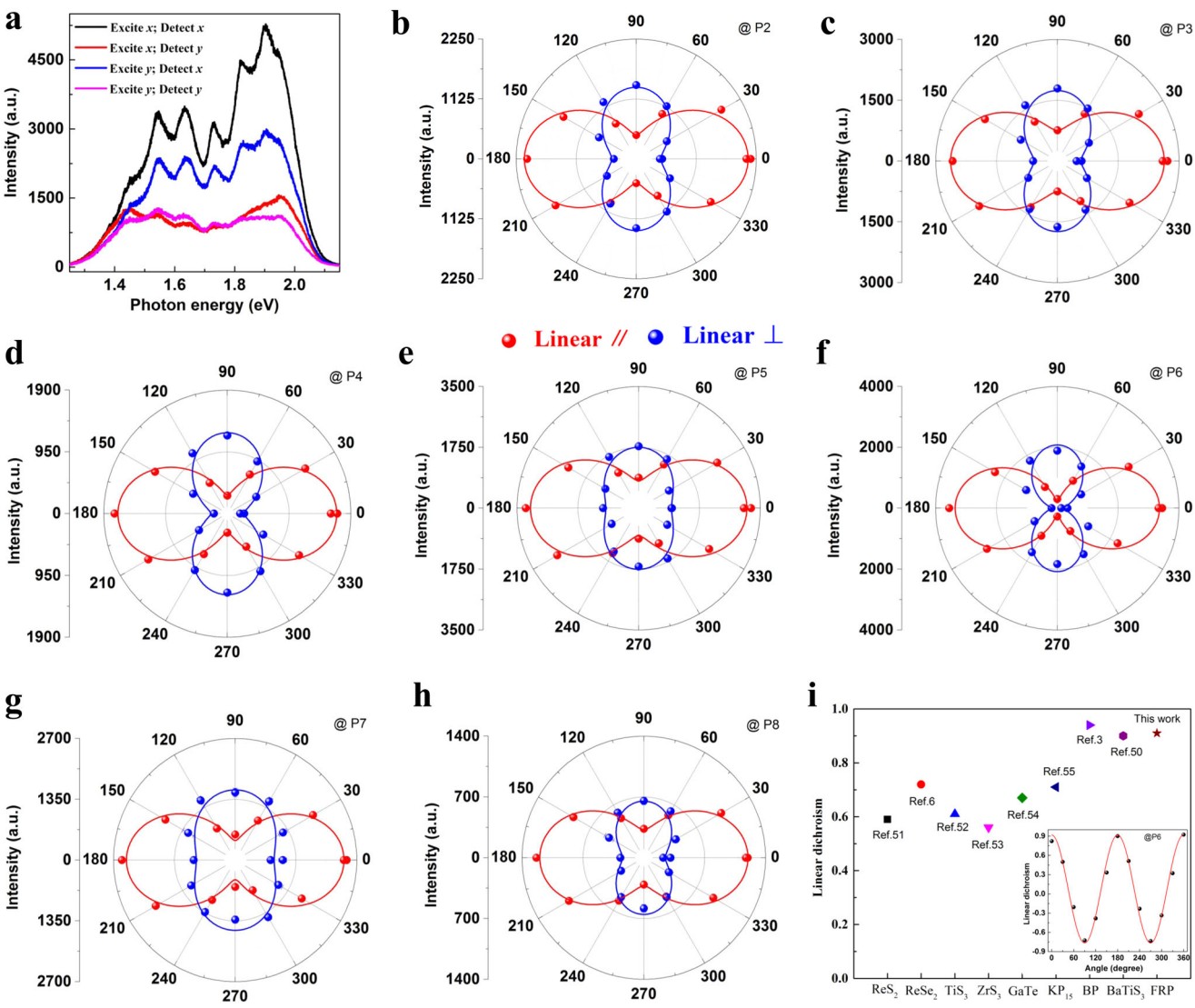

**Fig. 3 Highly anisotropic PL emission in 1D vdW FRP. a** PL spectra of a 1D vdW FRP flake under four different polarization configurations with the 532 nm excitation laser. The $x$ ($y$) axis is parallel (perpendicular) to the $b$ direction of 1D vdW FRP crystal. **b–h** Polar plots of emission intensity of P2–P8 as a function of polarization angle $\theta$ for copolarized (the polarizations of incident and emission light are parallel to each) and cross-polarized (the polarizations of incident and emission light are perpendicular to each) configurations. The red (blue) solid lines are fitted curves using a $\cos^2\theta$ ($\sin^2\theta$) function plus an offset. **i** The comparison of linear dichroism of FRP with the well-known quasi-1D van der Waals materials. The inset: the linear dichroism of P6 versus the polarization angle $\theta$.

(linearly parallel, red) and cross-polarized (linearly perpendicular, blue) configurations. Being akin to the linear PL response, the THG intensities in 1D vdW FRP are also strongly dependent on the polarization angle $\theta$ and possess the largest value when both the excitation and detection polarizations are aligned with the $b$ direction. The ratio between the brightest and weakest THG emission is larger than an order of magnitude, indicating the highly anisotropic third-order nonlinear optical response in 1D vdW FRP. Moreover, based on the evolution of THG intensities with polarization angle $\theta$, we can quantify the degree of THG anisotropy as $\rho = \frac{I_{co}(\text{THG}) - I_{\text{cross}}(\text{THG})}{I_{co}(\text{THG}) + I_{\text{cross}}(\text{THG})}$, where $I_{co}(\text{THG})$ and $I_{\text{cross}}(\text{THG})$ are the THG intensities detected co- and cross-polarized to the pump light polarization, respectively. The result shows that the degree of THG anisotropy can reach up to 86% when the excitation polarization is along the $b$ direction of the 1D vdW FRP crystal (Supplementary Fig. 12). The coexistence of the giant THG anisotropy and ultrastrong third-order nonlinear

response in 1D vdW FRP would pave the way toward a wealth of future integrated photonic and optoelectronic applications.

**Phonon anisotropies.** Finally, we demonstrate the anisotropic vibrational properties in 1D vdW FRP crystal. Figure 5a shows the nonpolarized Raman spectra at room temperature for an exfoliated 1D vdW FRP flake excited by a 2.33-eV laser in the confocal backscattering geometry. Interestingly, we can well distinguish up to 24 sharp phonon peaks by Lorentzian fitting scheme at ~ 108.7, 113.7, 143.3, 166, 179.1, 211.4, 241.3, 250.7, 278.4, 283.9, 296.5, 304.6, 354.4, 360.8, 369.6, 376.2, 391.7, 403.2, 410, 420.2, 431.4, 444.3, 454.2, and 472.2 cm$^{-1}$, sequentially labeled as R1–R24. The experimental phonon energies are in good line with the theoretical values based on DFT calculations (inset of Fig. 5a). The observation of so many Raman modes in 1D vdW FRP benefits from the low crystal symmetry and a variety of atoms in the unit cell. To unravel the anisotropic

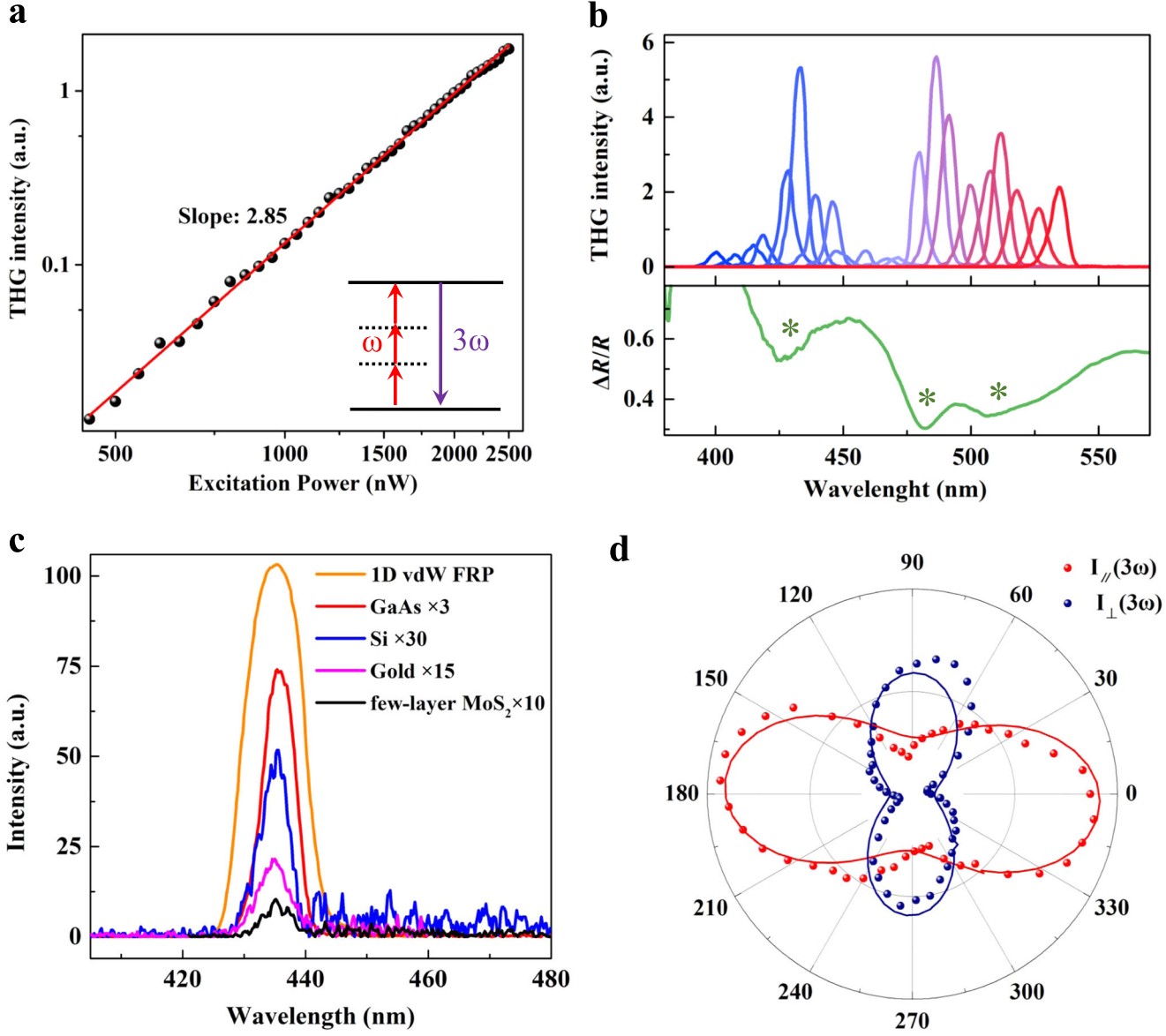

**Fig. 4 Ultrastrong and anisotropic third-order nonlinear optical response in 1D vdW FRP. a** THG intensities as a function of the excitation power with a fundamental pulse wavelength of ~ 1300 nm. The inset is the schematic diagram for the THG process. **b** Upper panel: THG spectra as a function of generated photon energies. Lower panel: the reflectance contrast spectrum $\frac{\Delta R}{R}$ of 1D vdW FRP, $\frac{\Delta R}{R} = \frac{R_{sample} - R_{substrate}}{R_{substrate}}$, where $R_{sample}$ and $R_{substrate}$ are the reflected spectra measured from the sample and substrate, respectively. **c** Comparison between the THG intensity of 1D vdW FRP and other well-known materials (e.g., GaAs, Si, gold, and few-layer MoS₂) with large third-order nonlinear optical responses under the same excitation conditions. The THG intensities of GaAs, Si, gold and few-layer MoS₂ thin films are multiplied by a factor of 3, 30, 15, and 10, respectively, for comparison. **d** Polar plots of THG intensity versus the polarization angle $\theta$ for co-polarized (red) and cross-polarized (blue) configurations.

vibrational properties, we perform the angle-resolved polarized Raman spectroscopy on exfoliated 1D vdW FRP flakes under copolarized and cross-polarized configurations (see Methods for details). In line with our expectations from the 1D vdW structure, all the phonon modes show strong anisotropy and are strongly crystalline orientation dependent, exhibiting 2-lobed/4-lobed and 4-lobed shapes for copolarized and cross-polarized configurations, respectively (Fig. 5b–i and Supplementary Fig. 13).

Within the Placzek approximation, the Raman scattering intensity can be expressed by the second-rank Raman tensor $(R)$ as $I \propto |e_i \cdot R \cdot e_s|^2$, where $e_i$ and $e_s$ are the unit polarization vectors of the incident and scattered light, respectively[15,59]. For 1D vdW FRP crystal, it crystallizes in a triclinic structure and belongs to the $C_i$ point group (space group $P\bar{1}$, No. 2)[35]. In the light of group theory, there is only one type of irreducible

representation $(A_g)$ for phonon modes at the center of BZ and the corresponding Raman tensor is described as[55,60]:

$$R(A_g) = \begin{pmatrix} a & d & e \\ d & b & f \\ e & f & c \end{pmatrix}.$$ Then, the $\varphi$-dependent Raman scatter-

ing intensities for copolarized and cross-polarized configurations can be described as follows:

$$I(\|) \propto \left| a\cos^2\varphi + b\sin^2\varphi + d\sin2\varphi \right|^2 \qquad (2)$$

$$I(\perp) \propto \left| \frac{(a-b)}{2}\sin2\varphi - d\cos2\varphi \right|^2 \qquad (3)$$

where $\varphi$ is the angle between the polarization of the incident laser and the [1,0,0] direction of 1D vdW FRP flake. As shown by the

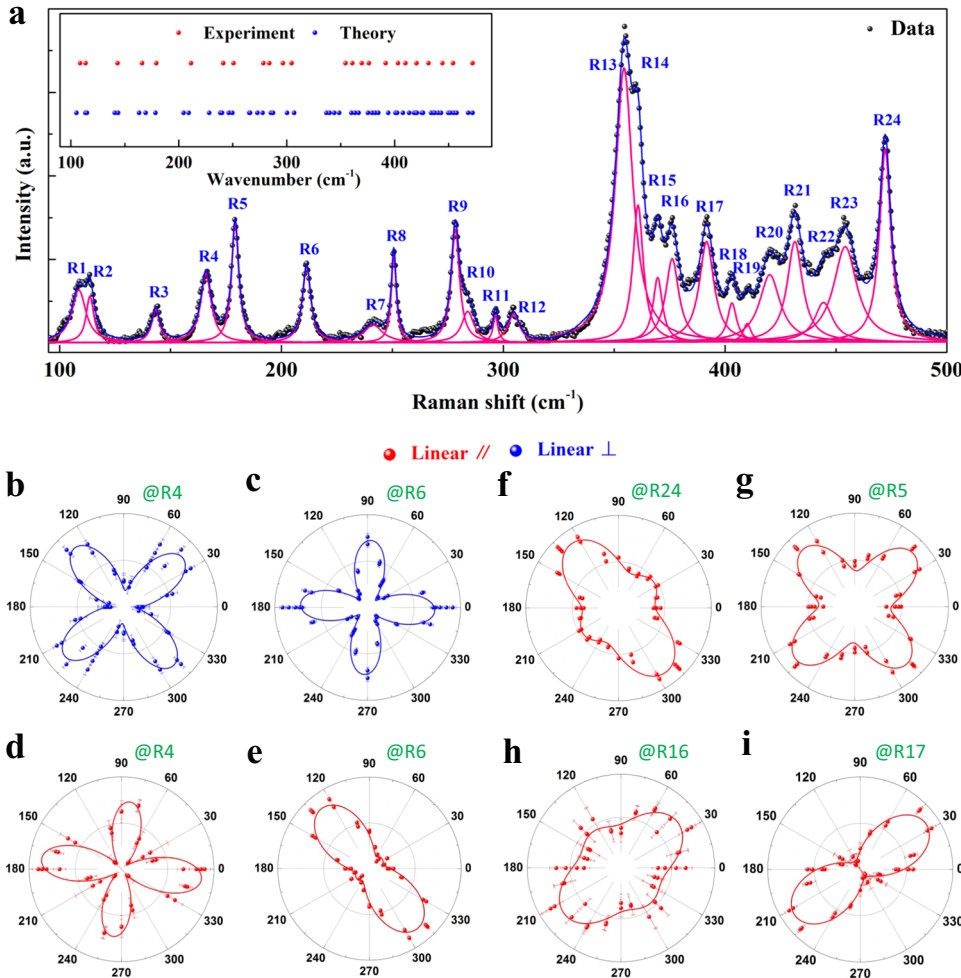

**Fig. 5 Giant anisotropic lattice vibration properties in 1D vdW FRP. a** Nonpolarized Raman spectra (black dots) for the exfoliated 1D vdW FRP flake, excited by a 2.33-eV excitation laser at room temperature. The Raman spectra of the 1D vdW FRP flake are fit to a sum of Lorentzians (blue and pink curves), from which we can obtain 24 sharp phonon peaks, labeled as R1–R24. The inset is the comparison between experimental (red) and calculated phonon energies (blue). **b–i** Polar plots of Raman intensity for six representative phonons as a function of polarization angle $\theta$ under copolarized (red dots) and cross-polarized (blue dots) configurations. The red and blue solid lines are fitted curves using Eqs. (2) and (3), respectively.

solid lines in Fig. 5b–i and Supplementary Fig. 13, Eqs. (2) and (3) can perfectly describe the experimental data under copolarized or cross-polarized configurations, respectively.

Although there is only one kind of phonon mode $A_g$, the evolution of Raman intensity with $\varphi$ can exhibit various features, depending on the ratio between different matrix elements in Raman tensor. Overall, the angle $\varphi$-dependent phonon intensity can be divided into two categories, according to the angle $\varphi_{max}$ corresponding to the maximum Raman intensity under cross-polarized configuration: type A with $\varphi_{max} = 45°$, 135°, 225°, and 315° and type B with $\varphi_{max} = 0°$, 90°, 180°, and 270°. Among the observed 24 phonons, type A includes only four phonon modes (e.g., R4 in the Fig. 5b), and the remaining phonon modes belong to type B (Supplementary Fig. 13). Based on the ratio between intensity at $\varphi = 135°/315°$ and $\varphi = 45°/225°$ in copolarized configuration, type B can be further divided into five categories. For example, the angle $\varphi$ dependence of the Raman intensities under parallel polarization configurations can show 2-lobed shapes with $\varphi_{max} = 135°/315°$ (e.g., R6 in the Fig. 5e) and 45°/225° (e.g., R17 in Fig. 5i), and exhibit 4-lobed shapes with a ratio of >1 (e.g., R24, Fig. 5f), ~1 (e.g., R5, Fig. 5g) and <1 (e.g., R16, Fig. 5h) between intensity at $\varphi = 135°/315°$ and $\varphi = 45°/225°$. For

other phonon modes, $\varphi$-dependent intensities follow one of the polar plots in Fig. 5b–i (Supplementary Fig. 13).

In conclusion, we report the FRP, a member of 1D vdW materials, through both the experimental measurements and theoretical calculations. Via angle- and polarization-resolved measurements of PL, THG, and Raman scattering, we reveal that the 1D vdW FRP harbors an unusually high anisotropy in both linear and nonlinear optical responses. The degree of PL and THG anisotropy can reach ~90% and 86%, respectively, comparable to the current record values. More significantly, our results uncover that 1D vdW FRP belongs to an exotic semiconductor in the visible range and possesses strong PL emission intensity and third-order nonlinear optical responses. The simultaneous realization of giant linear/nonlinear anisotropy and large optical responses in 1D vdW FRP would pave the way toward the engineering and development of advanced technological applications in nanophotonics, nanoelectronics, and optoelectronics.

## Methods
**Sample preparation.** 1D vdW FRP bulk crystals are synthesized by flux zone growth technology (2D Semiconductors). Initially, yellow phosphorus as the precursors (99.9999% purity) and Si powder as the catalysts (99.9999% purity) are

sealed in a quartz ampoule under $10^{-6}$ Torr pressures. The ampoule is annealed at high temperatures 900 °C with the thermal drop around 50 °C for three weeks to synthesize the crystals. Then the ampoule is cooled down to room temperature by natural cooling technique (1 °C/min). 1D vdW FRP flakes are mechanically exfoliated by scotch tape from the synthesized bulk crystal. In close analogy to the established processes for 2D vdW materials (e.g., graphene, black phosphorus, and $MoS_2$), we use 300-nm $SiO_2/Si$ and ordinary adhesive tape as the substrate and transfer medium, respectively. To remove the ambient adsorbates and enhance the adhesion force, the $SiO_2/Si$ substrate is ultrasonically cleaned in acetone, 2-propanol, and deionized water, and then subjected to oxygen plasma before the process of mechanical exfoliation.

**Sample characterizations.** The powder X-ray diffraction patterns are collected with a Rigaku SmartLab X-ray diffractometer in transmission mode using Cu-Kα radiation in the 2θ range of 10–70° with a scan-step width of 0.02°. The theoretical simulation of XRD patterns is calculated by Mercury, a free software designed by Cambridge Crystallographic Data Centre (https://www.ccdc.cam.ac.uk/Community/csd-community/freemercury/). The XPS measurements are made using a Kratos Axis Ultra system, equipped with a monochromatic AlKα X-ray source. The diameter of the spot size is about 110 μm, and a charge neutralizer is used to correct surface charging. HRTEM imaging is carried out on an aberration-corrected JEOL 2200FS transmission electron microscope operated at 200 keV. The chemical composition and element distribution are determined by energy dispersive X-ray spectroscopy attached to the transmission electron microscope. The AFM images are performed by Asylum Research Cypher S under the tapping mode with the AC160TS tip at room temperature under ambient conditions.

**PL and Raman measurements.** PL and Raman spectra are acquired using a micro-Raman spectrometer (Horiba LabRAM HR Evolution) in a confocal backscattering geometry (confocal pinhole of 100 μm). A solid-state laser at 532 nm is focused onto the samples along the z direction by a ×100 objective with a spot size less than 1 μm. The backscattered signal is collected by the same objective and dispersed by a 600-groove $mm^{-1}$ grating for PL measurement and a 1800-groove $mm^{-1}$ grating to achieve Raman spectral resolution better than 1 $cm^{-1}$. The laser power during PL and Raman measurement is kept below 100 μW in order to avoid sample damage and excessive heating. The integration time is 5 s. To uncover the fascinating anisotropy, we perform the angle- and polarization-resolved measurements. The excitation laser beam is passed through a linear polarizer and then a half-wave plate. The half-wave plate can tune the polarization angle in the xy plane. The backscattered light is passed through the same half-wave plate. Another linear polarizer is placed in front of the nitrogen-cooled CCD to selectively detect the components parallel/perpendicular to the polarization of excitation laser, referred to as the co-/cross-polarized configuration. In time-resolved PL measurements, the excitation beam is produced by a mode-locked Ti:sapphire laser with a wavelength of 400 nm, a pulse-repetition rate of 84 MHz, and a pulse duration of less than 100 fs. The beam diameter on the sample is ~2 μm. A streak camera (Hamamatsu) with a nominal time resolution of 1 ps is used to measure the time-resolved PL spectra. The streak camera is operated at a moderate gain, optimized for the best signal-to-noise ratio, with a 100-s integration time for each spectrum.

**THG measurements.** Femtosecond (~200-fs) pulses with wavelengths ranging from 1200 to 1600 nm are generated by a TOPAS system (Light conversion) pumped with an amplified Ti:sapphire laser at a repetition rate of 2 kHz. The pulsed laser is then aligned into a homemade microscopy system with a ×40 objective (NA. 0.75) and vertically illuminates the sample. The full width at half-maximum of the beam spot is ~2.5 μm. The generated THG signals are collected by the same objective and then focused by a lens into a monochromator with a photomultiplier-tube (PMT) detector. The short-pass filters are used to get rid of the residual fundamental beam. The powers of incident pulses at different wavelengths are measured behind the objective with a calibrated power meter (Thorlabs S148C). The powers of detected THG signals are calibrated by a standard white light. For the polarization measurements, we rotate the sample while keeping the polarization of incident light fixed. The THG signal with polarization parallel or perpendicular to the polarization of incident lights is detected.

**First-principles calculations.** DFT calculations are carried out using the generalized-gradient approximation for the exchange-correlation potential, the projector-augmented wave method[61] and a plane-wave basis set as implemented in the Vienna ab-initio simulation package (VASP)[62] and Quantum Espresso (QE)[63]. Dispersion correction is made at the van der Waals density-functional (vdW-DF) level[64], with the optB88 functional for the exchange potential (optB88-vdW)[63,64]. Note that the adopted optB88-vdW DFT functional, which introduces a nonlocal correlation to the standard GGA functional for capturing the dispersion interactions, has been proven to be accurate in describing the geometric-structure-related properties and the type of bandgaps (e.g., direct vs. indirect) of vdW materials[8,26]. Other functionals, including optB86b-vdW, PBE-D3, and modified Becke–Johnson (mBJ), were used for comparison and verification. Their results are available in the Supplementary Information. The shape and volume of the FRP bulk crystal are fully relaxed, until the residual force per atom is less than 0.01 eV/Å. The starting geometry is constructed based on the previous powder XRD measurements of ref. [35], downloaded from Inorganic Crystal Structure Database. The kinetic energy cutoff for the plane-wave basis set is set to 700 eV for geometric optimization and 400 eV for electronic structure calculation. Two k-meshes of $5 \times 5 \times 5$ and $3 \times 5 \times 5$ are adopted to sample the first BZ of FRP in structural optimization and electronic structure calculation, respectively. A denser k-point mesh of $13 \times 21 \times 21$ is utilized to plot the Fermi surface. Density-functional perturbation theory[65] is employed to calculate Raman activity and frequencies of vibrational modes at the Γ point using the Quantum Espresso package. All phonon-related properties are calculated using a $3 \times 3 \times 3$ k-mesh and a single q point with a plane-wave energy cutoff of 50 Ry.

## Data availability
The data that support the findings of this study are available from the corresponding authors on a reasonable request.

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

## Acknowledgements

The authors acknowledge support from Academy of Finland (Grant Nos. 333099, 314810, 333982, 336144, and 336818), Academy of Finland Flagship Programme (Grant No. 320167, PREIN), the European Union's Horizon 2020 research and innovation program (Grant Nos. 820423, S2QUIP; 965124, FEMTOCHIP), the EU H2020-MSCA-RISE-872049 (IPN-Bio), and ERC (Grant No. 834742). W.J. gratefully acknowledges financial support from the Ministry of Science and Technology (MOST) of China (Grant No. 2018YFE0202700), the National Natural Science Foundation of China (Grants Nos. 11622437, 61674171 and 11974422), the Strategic Priority Research Program of Chinese Academy of Sciences (Grant No. XDB30000000), the Fundamental Research Funds for the Central Universities, China, and the Research Funds of Renmin University of China [Grant Nos. 16XNLQ01, 19XNQ025 (W.J.), and 19XNH065 (X.Z.)]. Calculations are performed at the Physics Lab of High-Performance Computing of Renmin University of China and Shanghai Supercomputer Center. We also acknowledge the provision of facilities of Aalto University at Micronova and OtaNano-Nanomicroscopy Center.

## Author contributions

L.D., W.J. and Z.S. supervise the research; L.D. conceives this project, analyzes the data, and performs the Raman measurements; Y.Z. performs the PL characterization under the supervision of G.Z.; X.H. and Y.W. perform the THG measurements; L.W. J.Q., and W.J. perform theoretical calculations; X.H., L.Y., X.L. and X.B. perform the TEM character-ization; X.Y.B. performs the XRD characterization and simulation. X.H. and J.L. perform the XPS characterization. Y.D. and U.M.G. perform the time-resolved PL measurements. L.D., L.W., W.J. and Z.S. write the paper. All authors comment on the paper.

## Competing interests

The authors declare no competing interests.
