## [Peer Review File · Nature Communications]

REVIEWER COMMENTS

Reviewer #1 (Remarks to the Author):

In this submission the authors report a comprehensive and fascinating study of the structure, linear and non-linear optics and vibrational properties of a new allotrope of phosphor, FRP, which has an interesting crystal structure featuring 1D tubes along the b axis. This is a very detailed and mostly well-reasoned study of a number of interesting features of this new compound, which would certainly be of broad interest.

The angle- and polarisation-resolved optical studies show convincingly that this system has a very large anisotropy, as expected from the crystal structure.

Overall I think this is a really interesting set of results that are worth considering further for publication in Nature Communications. But as it is currently phrased in the abstract and conclusions, the manuscript makes the claims about FRP, e.g. that it "possesses ultra-strong PL emission intensity and third-order nonlinear optical responses", that I am not convinced are fully evidenced.

For instance, these assertions about the strength appear to be based on Figures 2b and Figures 4c, where the signal from an exfoliated FRP sample is far stronger than that of monolayer MoS₂. But this is not a fair comparison, as the exfoliated FRP samples are at least 40nm thick (line 133), and hence the sample is far thicker than a monolayer of MoS₂. If the authors want to compare the relative strength, then a better figure of merit would be e.g. PL counts per incident photon absorbed by the sample, or THG intensity divided by sample thickness. In that case I think the relative strength of the PL and THG would be much closer to that of MoS₂, and certainly not 40x higher as reported in the current manuscript.

Apart from this major concern I have a few more minor comments (below). I think that after addressing the above and below concerns the paper can certainly be considered again for publication in Nature Comms.

1. I think the title is a bit too general and should be made more specific to mention the FRP material system. 1D vdW semiconductor might mean one of a variety of materials, as mentioned in the introduction.
2. In the introduction the authors state that "up to now, only a few 1D vdW materials (e.g., elemental Te_{24,25} and Sb_{2Se327}) have been experimentally uncovered, mainly focusing on the electrical transport properties". The authors should cite other recent experimental work on 1D vdW materials containing MoS₂, BN and carbon nanotubes (Burdanova et al., Nano Letters 2020, dx.doi.org/10.1021/acs.nanolett.0c00504 and Rong et al., Science 2020, dx.doi.org/10.1126/science.aaz2570) that has also looked at some of the optical properties, but not the anisotropic and non-linear optics as in the authors' present work.
3. Lines 145-168 - The authors show PL spectra with around 8 strong resonances, which they describe as excitons. However the proof that these are from excitons rather than interband transitions is not clear. Just seeing a resonance in the PL does not mean it results from an exciton – as the authors mention, the equal energy spacing of the peaks is consistent with phonon sidebands such as is often seen in PL emission in direct gap bulk semiconductors where there is strong electron-phonon coupling (e.g. ZnO).
4. Lines 160-161 – In the discussion of the high density of states, would it make sense to draw an analogy here to the well-known van Hove singularity in 1D materials (e.g. carbon nanotubes)?
5. Line 164 – Is this really an indirect semiconductor? DFT is well-known to not get bandstructure exactly right – for instance the bandgap is calculated to be 1.57eV and indirect, but the PL emission is at 1.8eV. Perhaps the PL emission is between states that are higher in the CB and VB? For instance, there is still PL emission from bilayer MoS₂ across the K-point bandgap, despite the indirect nature of the bandgap, it is just a lot weaker than for monolayer MoS₂.
6. Regarding absorption, the authors show differential reflectance (Figure 4 b) from experiment in the 400-

550nm range, and absorbance calculated from DFT (Figure S6). But what about experimental reflectance or absorption in the 1.4eV to 2eV range, covering the claimed indirect bandgap at 1.57eV?

7. The authors refer also to three "exciton" resonances at about 2.86eV, 2.55eV and 2.43eV. But these are really just resonances in the absorption spectra, arising from poles in the dielectric function. It is natural to expect that the THG intensity would be resonantly enhanced when the third harmonic is resonant to a pole in the response function. But there is no evidence that the poles arise from excitons – indeed "interband transitions" would be much better terminology than excitons. Just because you have a peak in the absorption spectrum does not mean it is an exciton! The interband transition rate (controlling the absorption) is strongly enhanced near energies where there is a large matrix element.

8. Indeed, perhaps the authors can comment (for the non-expert in DFT!) on what physics the optB88-vdW DFT functional actually captures? Does it reproduce excitonic absorption peaks?

9. 180 – the anisotropy is certainly evidenced by the PL anisotropy. But again, the same argument can be made for interband transitions between (uncorrelated) electron hole pairs.

10. 198 – the phrase "linear exciton process" is not quite precise enough. Excitons themselves are not a "process", they are a quasiparticle. Perhaps "strong in-plane anisotropy of the PL emission" would be better.

James Lloyd-Hughes

Reviewer #2 (Remarks to the Author):

In the manuscript "Giant anisotropic photonics in 1D van der Waals semiconductor" LuoJun Du et al. present an study of the photoluminescence, third harmonic generation and Raman spectroscopy of fibers of red phosphorus. The authors find a remarkable anisotropic light emission and third harmonic generation. The manuscript is interesting as more and more part of the community working on 2d materials are getting interested by quasi-1D 2D materials. Fibrous red phosphorus is one example, together with Tellurium or Sb₂Se₃, of 'real 1D' van der Waals material.

The results are well-presented, the text is clear and the statements made by the authors are sufficiently hold by their results. I missed a bit of more direct comparison with other quasi-1D van der Waals materials (like BP, ReS₂, ReSe₂, TiS₃, ZnSe₃, SnS, SnSe, etc...). This would make the manuscript more appealing to the community.

Regarding the thickness of the flakes. Figure 2a is not clear enough to claim that the thinnest flake has 40nm. It is not clear that the lower-right corner of the image is the substrate and not another part of the flake. Could the authors provide a optical image of that region as well?

It seems that it is difficult to obtain thin ribbons of red phosphorus, the authors should discuss this point to make it clearer to the community. Would liquid phase exfoliation help to get thinner ribbons/fibers?

In the main text it seems that the material has been synthesized in-house but then in the materials section it is said that has been purchased to 2D Semiconductors. Please revise the text to avoid confusion of the readership.

In view of this, I believe that the manuscript could be accepted for publication after addressing these minor points.

Reviewer #3 (Remarks to the Author):

The report highly anisotropic optical properties of fibrous red phosphorus (FRP). This is a 1D vdW material, so there is structural motivation for the proposed study. The presented material needs more careful analysis and not in publishable form. I list my concerns below:

1. The authors claim intense PL emission 40x more than monolayer MoS₂. This statement has no meaning. What is the quantum efficiency of PL emission in FRP? It will be better to make quantitative claims rather than misleading comparisons with monolayer MoS₂.
2. What is the exciton life-time in this material? Does the inefficient indirect gap emission still manifest with high efficiency? What is the physical reason behind this effect?
3. What is the Debye temperature of this material? Given the complex structure, I assume that the phonons may be populated at room temperature. So, I am even more surprised at the exciton-phonon interaction as the cause for the ordered PL peaks.
4. The polarized PL studies show a reduction in intensity, but the broadband emission seems to remain for all conditions. How does this make sense? Can you measure these at low temperatures to see if some of the peaks become sharper. If the exciton-phonon interaction is critical, at low temperatures only the direct emission should be observed.
5. The comparison of third order non-linear optical response to common materials such as gold, GaAs, MoS₂ is misleading. It will be good to compare it with materials that are typically known for this property. It is not necessary that FRP should outperform them, but that is a fair comparison. In these cases, it is beneficial to quantify their performance in terms of a susceptibility coefficient.
6. The polarized Raman study is interesting, but it is unclear how that helps us understand the polarized response of the PL study. I see that the symmetry of the Raman response studied for a few peaks may not correspond to the polarized PL response.

Overall, I feel that the results presented could be interesting, but the analysis presented is poor. It doesn't help the reader understand the importance of the work. I would not recommend the publication of this article even in a specialized journal.

Point-by-Point Responses to the Reviewers' Comments

We thank all three reviewers for their very keen interest in our manuscript. Reviewer #1 comments “*Overall I think this is a really interesting set of results that are worth considering further for publication in Nature Communications*”; Reviewer #2 comments “*I believe that the manuscript could be accepted for publication after addressing these minor points*” and Reviewer #3 comments “*the results presented could be interesting*”.

We also appreciate the reviewers' insightful and constructive comments that help to improve our manuscript. We have addressed all of them in the revised version of our manuscript and in the replies (in blue). We also include a list of changes (in red) made in the main manuscript.

Reviewer # 1 (Remarks to the Author):

In this submission the authors report a comprehensive and fascinating study of the structure, linear and non-linear optics and vibrational properties of a new allotrope of phosphor, FRP, which has an interesting crystal structure featuring 1D tubes along the b axis. This is a very detailed and mostly well-reasoned study of a number of interesting features of this new compound, which would certainly be of broad interest.

The angle- and polarisation-resolved optical studies show convincingly that this system has a very large anisotropy, as expected from the crystal structure.

Overall I think this is a really interesting set of results that are worth considering further for publication in Nature Communications. But as it is currently phrased in the abstract and conclusions, the manuscript makes the claims about FRP, e.g. that it “possesses ultra-strong PL emission intensity and third-order nonlinear optical responses”, that I am not convinced are fully evidenced.

For instance, these assertions about the strength appear to be based on Figures 2b and Figures 4c, where the signal from an exfoliated FRP sample is far stronger than that of monolayer MoS₂. But this is not a fair comparison, as the exfoliated FRP samples are at least 40 nm thick (line 133), and hence the sample is far thicker than a monolayer of MoS₂. If the authors want to compare the relative strength, then a better figure of merit would be e.g. PL counts per incident photon absorbed by the sample, or THG intensity divided by sample thickness. In that case I think the relative strength of the PL and THG would be much closer to that of MoS₂, and certainly not 40x higher as reported in the current manuscript.

Response 1

We thank the reviewer for his positive assessment of our work and insightful comments for improvement.

We fully agree with the reviewer that owing to the different thicknesses, it is better to compare the PL quantum efficiency (e.g. PL counts per incident photon absorbed by the sample) between monolayer MoS₂ and the exfoliated FRP sample. Considering that monolayer MoS₂ possesses an absorbance of ~10% under 2.33 eV excitation [e.g., *Nano Lett.*

13, 3664 (2013)], the PL quantum efficiency of FRP should be more than four times larger than that of monolayer MoS₂, even if we assume that the absorption of FRP is 100%.

In order to better compare the THG between FRP and MoS₂, we derive the THG third-order susceptibility $\chi^{(3)}$:

$$\chi^{(3)} = \frac{4\varepsilon_0 c^2}{3\omega d} \sqrt{n_\omega^3 n_{3\omega} \frac{I_{3\omega}}{I_\omega^3}} \quad (\text{R1})$$

where ε_0 , c and d are the permittivity of vacuum, speed of light and sample thickness, respectively [e.g., *Phys. Rev. B* 87, 121406(R) (2013)]. And n_ω ($n_{3\omega}$) and I_ω ($I_{3\omega}$) are the refractive index at frequency ω (3ω) and pump (THG) intensity, respectively. Based on Equation (R1), we obtain the THG third-order susceptibility of FRP: $\chi^{(3)}(\text{FRP}) = 2.67 \times 10^{-18} \text{m}^2 \text{V}^{-2}$, which is three times larger than that of MoS₂ ($\chi^{(3)}(\text{MoS}_2) = 8.99 \times 10^{-19} \text{m}^2 \text{V}^{-2}$). Note that calculated refractive indexes n_ω and $n_{3\omega}$ are used to derive the THG third-order susceptibility of FRP because of the absence of experimental results. Figure R1 shows the wavelength dependent refractive index of FRP obtained by density functional theory calculations. For the pump wavelength of 1300 nm used in Figure 4c of the main text, n_ω and $n_{3\omega}$ are 3.11 and 3.98, respectively.

Figure R1. Wavelength dependent refractive index of FRP obtained by theory calculations.

To address this comment, we have made the following changes in the revised manuscript and supplementary information:

1) We have added a sentence to discuss the PL quantum efficiency in the revised manuscript (page 6, lines 150-153).

2) We have added the discussions about the THG third-order susceptibility and Equation (R1) into the main text of the revised manuscript (page 9, lines 244-253).

3) Figure R1 has been added into the revised supplementary information as Figure S7 (page 4).

4) To be more rigorous, we have changed the word that describes the PL/THG intensity. For example, “ultra-strong” and “extremely large”/“unusually large” have been changed to “strong” and “large”, respectively.

Apart from this major concern I have a few more minor comments (below). I think that after addressing the above and below concerns the paper can certainly be considered again for publication in Nature Comms.

1. I think the title is a bit too general and should be made more specific to mention the FRP material system. 1D vdW semiconductor might mean one of a variety of materials, as mentioned in the introduction.

Response 2

We thank the reviewer for the comments. We have changed the title to “*Giant anisotropic photonics in 1D van der Waals semiconductor fibrous red phosphorus*”.

2. In the introduction the authors state that “up to now, only a few 1D vdW materials (e.g., elemental $Te^{24,25}$ and $Sb_2Se_3^{27}$) have been experimentally uncovered, mainly focusing on the electrical transport properties”. The authors should cite other recent experimental work on 1D vdW materials containing MoS_2 , BN and carbon nanotubes (Burdanova et al., *Nano Letters* 2020, [dx.doi.org/10.1021/acs.nanolett.0c00504](https://doi.org/10.1021/acs.nanolett.0c00504) and Rong et al., *Science* 2020, [dx.doi.org/10.1126/science.aaz2570](https://doi.org/10.1126/science.aaz2570)) that has also looked at some of the optical properties, but not the anisotropic and non-linear optics as in the authors' present work.

Response 3

We thank the reviewer for pointing out these two important prior works. We have cited them in the introduction part of the revised manuscript [Ref. 32: *Nano Lett.* 20, 3560-3567 (2020) and Ref. 33: *Science* 367, 537-542 (2020)]. In addition, we have cited another nice paper that shows the prominent intertube excitonic effects of 1D vdW materials containing MoS_2 , BN and carbon nanotubes [Ref. 34: arXiv:2104.09430 (2021)].

3. Lines 145-168 - The authors show PL spectra with around 8 strong resonances, which they describe as excitons. However the proof that these are from excitons rather than interband transitions is not clear. Just seeing a resonance in the PL does not mean it results from an exciton – as the authors mention, the equal energy spacing of the peaks is consistent with phonon sidebands such as is often seen in PL emission in direct gap bulk semiconductors where there is strong electron-phonon coupling (e.g. ZnO).

Response 4

We thank the reviewer for the comments. As suggested by the reviewer, we have changed “exciton peaks”, “exciton emission intensity” and “exciton anisotropy” to “PL peaks”, “PL emission intensity” and “PL anisotropy” in the revised manuscript, respectively.

4. Lines 160-161 – In the discussion of the high density of states, would it make sense to draw an analogy here to the well-known van Hove singularity in 1D materials (e.g. carbon nanotubes)?

Response 5

Yes, we agree with the reviewer that the high density of states in FRP can be viewed as an analogy to the van Hove singularity in 1D materials (e.g. carbon nanotubes).

To address this comment, we have changed the sentence “..... indicating the quasi-flat band with a high density of states.” to “..... indicating the quasi-flat band with a high density of states which can be viewed as an analogy to the van Hove singularity in 1D materials.” in the revised manuscript (page 7).

5. Line 164 – Is this really an indirect semiconductor? DFT is well-known to not get band structure exactly right – for instance the bandgap is calculated to be 1.57 eV and indirect, but the PL emission is at 1.8 eV. Perhaps the PL emission is between states that are higher in the CB and VB? For instance, there is still PL emission from bilayer MoS₂ across the K-point bandgap, despite the indirect nature of the bandgap, it is just a lot weaker than for monolayer MoS₂.

Response 6

We fully agree with the reviewer that standard DFT calculations usually underestimate the bandgaps of semiconductors, primarily because of the over-estimated delocalization of electrons and continuous exchange-correlation functionals. However, DFT calculations often perform very well in telling us whether the bandgap of a semiconductor is direct or indirect. For example, while the standard DFT reveals a bandgap of ~1.4 eV for monolayer WS₂ (Figure R2a), which is more than 0.5 eV lower than the observed optical bandgap of 1.96 eV, it accurately predicts a direct to indirect bandgap transition from monolayer to bilayer/trilayer WS₂ [Figure R2, adapted from *Phys. Rev. B* **100**, 161404(R) (2019)].

To be more careful, we have calculated the band structures of FRP with four different exchange-correlation functionals, i.e., optB88-vdW (Figure R3a, also available in the main text), optB86b-vdW (Figure R3b), PBE-D3 (Figure R3c) and a meta-GGA, modified Becke-Johnson functional (mBJ, Figs. R3d-R3f). Note that since the mBJ functional doesn't work in crystal structure optimization, the band structures in Figs. R3d, R3e and R3f were obtained using the optimized crystal structures in Figs. R3a, R3b and R3c, respectively. We may have to emphasize that the mBJ functional shall predict a reliable fundamental bandgap because the errors from over-estimated delocalization of electrons and/or holes are corrected in the calculations [e.g., *Phys. Rev. Lett.* 102, 226401 (2009)]. Indeed, the mBJ functional yields an indirect fundamental bandgap of 1.80-2.11 eV (Figs. R3d-R3f), which is very close to the energy of the P8 PL peak that we observed (2.05 eV, Figure 2b in the main text). Nevertheless, all calculation results show that FRP is an indirect semiconductor.

We are grateful to the reviewer for raising the comments about the origin of the PL emission. After seriously considering the possibility mentioned by the reviewer, we found that our data still prefer the indirect origin. We explain our considerations as follows. First, as we mentioned above, our PL experimental results are well consistent with the reliable value of indirect bandgap predicted by the mBJ functional. Second, if the observed PL emissions in FRP are from the direct transitions between higher states in the conduction band (CB) and valence band (VB), we should also observe PL emissions with lower energies from the indirect transitions between states in conduction band minimum (CBM) and valence band maximum (VBM). Although multiple PL peaks are observed, the equal energy spacing indicates that these peaks should have a common origin. In other words, we do not observe the indirect transitions with lower energy between states in CBM and VBM, if those observed multiple PL peaks of FRP belong to direct transitions between higher states in the CB and VB. In addition, for PL emission from the direct transitions, it generally appears as an individual single peak dominated by a “zero-phonon” line. In light of all these reasons, we believe that the experimentally observed multiple PL peaks in FRP, most likely, origin from the indirect transitions. It is worth stressing that the origin for the multiple PL peaks of FRP

deserves further in-depth studies (e.g., angle-resolved photoemission spectroscopy), which is beyond the scope of the present manuscript.

To address this comment, we have added the following sentences in the revised manuscript (page 6, lines 170-174):

“Note that the adopted optB88-vdW functional usually underestimates the bandgap of a semiconductor, primarily due to the over-estimated delocalization of electrons. The modified Becke-Johnson (mBJ) functional with such issue corrected shall predict a reliable fundamental bandgap⁴⁷. Indeed, the mBJ functional yields an indirect fundamental bandgap of 1.80-2.11 eV for FRP (Supplementary Figure S8), which is very close to the energy of the P8 peak (2.05 eV, Figure 2b).”

In addition, Figure R3 have been added into the revised supplementary information as Figure S8 (page 5).

Figure R2 [adapted from *Phys. Rev. B* **100**, 161404(R) (2019)]. Band structure of monolayer (a), bilayer (b) and trilayer (c) WS₂ obtained by DFT calculation.

Figure R3. Theoretical band structures of FRP with the optB88-vdW (a), optB86b-vdW (b), PBE-D3 (c) and modified Becke-Johnson (mBJ, d-f) functionals. Structures are optimized with the optB88-vdW (a, d), optB86b-vdW (b, e) and PBE-D3 (c, f) functionals, respectively.

All these band structures show that FRP is an indirect semiconductor. The mBJ functional shall predict a reliable fundamental bandgap because the errors from over-estimated delocalization of electrons and/or holes are corrected.

6. Regarding absorption, the authors show differential reflectance (Figure 4b) from experiment in the 400-550 nm range, and absorbance calculated from DFT (Figure S6). But what about experimental reflectance or absorption in the 1.4eV to 2eV range, covering the claimed indirect bandgap at 1.57eV?

Response 7

We thank the reviewer for the comments. Figure R4 shows the experimental reflectance contrast spectrum in the range of 380-900 nm. In the range of 1.3-2 eV (inset of Figure R4), the reflectance contrast spectrum is relatively smooth and no obvious absorption peak is observed.

To address this comment, Figure R4 has been added into the revised supplementary information as Figure S11 (page 6).

Figure R4. Reflectance contrast spectrum in the range of 380-900 nm. Inset is the reflectance contrast spectrum in the range of 1.3-2 eV.

7. The authors refer also to three “exciton” resonances at about 2.86eV, 2.55eV and 2.43eV. But these are really just resonances in the absorption spectra, arising from poles in the dielectric function. It is natural to expect that the THG intensity would be resonantly enhanced when the third harmonic is resonant to a pole in the response function. But there is no evidence that the poles arise from excitons – indeed “interband transitions” would be much better terminology than excitons. Just because you have a peak in the absorption spectrum does not mean it is an exciton! The interband transition rate (controlling the absorption) is strongly enhanced near energies where there is a large matrix element.

Response 8

We fully agree with the reviewer. As suggested by the reviewer, we have changed “exciton” to “interband transitions” in the revised manuscript.

8. Indeed, perhaps the authors can comment (for the non-expert in DFT!) on what physics the optB88-vdW DFT functional actually captures? Does it reproduce excitonic absorption peaks?

Response 9

We are grateful to the reviewer for raising the comments. The adopted optB88-vdW DFT functional [e.g., *Phys. Rev. B* 83, 195131 (2011)], which introduces a non-local correlation to the standard GGA functional for capturing the dispersion interactions, has been proven to be accurate in describing the geometric-structure-related properties and the type of bandgaps (e.g., direct versus indirect) of van der Waals materials [e.g., *Nanoscale* 8, 2740-2750 (2016) and *Sci. Bull.* 63, 159-168 (2018)]. Three additional functionals (e.g., optB86b-vdW, PBE-D3 and mBJ) have been used to verify the indirect bandgap nature of FRP obtained by the optB88-vdW functional.

In terms of the electronic structures and related optical properties, if the absorption peaks or PL emissions are from interband transitions without many-body electron-hole interactions, the calculated absorption peaks and/or fundamental bandgap obtained with the optB88-vdW functional should have lower energies than the experimental values. This is because that the optB88-vdW functional usually underestimates the bandgap of a semiconductor, as mentioned in our Response 6. For PL emissions, our experimental results (the energy of P8 peak, Figure 2b in the main text) are larger than the indirect bandgap predicted by the optB88-vdW functional, but match well the reliable value of indirect bandgap predicted by the mBJ functional. Note that for indirect interband transitions/phonon replicas, the highest PL emission energy (e.g., the energy of P8 peak, Figure 2b in the main text) can usually be regarded as the fundamental bandgap. This indicates that our PL emissions of FRP, most likely, originate from the interband transitions between uncorrelated or very weakly coupled electron hole pairs.

On the other hand, if the absorption peaks are from excitons, the calculated absorption spectra with the optB88-vdW functional are often comparable to experimental results since the excitonic effects can fortuitously cancel the well-known tendency of the optB88-vdW functional to underestimate the fundamental bandgap [e.g., *Phys. Rev. B* 94, 155428 (2016)]. In our theoretical absorption spectra obtained with the optB88-vdW functional [Supplementary Figure S6], three absorption peaks locate at ~ 2.37 eV, 2.61 eV and 2.89 eV, respectively, which are well consistent with the corresponding experimental values of ~ 2.43 eV, 2.55 eV and 2.86 eV [Figure 4b in the main text]. This comparison indicates that these three absorption peaks of FRP may come from excitons.

To address this comment, we have added the following sentences in the methods section of the revised manuscript (page 14):

“Note that the adopted optB88-vdW DFT functional, which introduces a non-local correlation to the standard GGA functional for capturing the dispersion interactions, has been proven to be accurate in describing the geometric-structure-related properties and the type of bandgap (e.g., direct versus indirect) of vdW materials^{8,26}. Other functionals, including optB86b-vdW, PBE-D3 and modified Becke-Johnson (mBJ), were used for comparison and verification. Their results are available in the Supplementary Information.”

In addition, we have added the following discussions in the revised supplementary information (page 4):

“Since the optB88-vdW DFT functional usually underestimates the bandgap of a semiconductor, the theoretical absorption peaks obtained using optB88-vdW should have lower energies than the experimental peaks, if those peaks originate from the interband transitions without or with very weak many-body electron-hole interactions. On the other hand, if the absorption peaks are from excitons, the calculated absorption spectra with the optB88-vdW functional are, most likely, comparable to experimental results since the excitonic effects can fortuitously cancel the underestimated bandgap that optB88-vdW functional usually shows¹. Our theoretical absorption spectra show three peaks located at ~ 2.37 eV, 2.61 eV and 2.89 eV, respectively, which are well consistent with the corresponding experimental values of ~ 2.43 eV, 2.55 eV and 2.86 eV [Figure 4b in the main text]. This comparison indicates that these three absorption peaks of FRP may come from excitons.”

9. 180 – *the anisotropy is certainly evidenced by the PL anisotropy. But again, the same argument can be made for interband transitions between (uncorrelated) electron hole pairs.*

Response 10

We thank the reviewer for the comments. We have changed “exciton anisotropy” to “PL anisotropy” in the revised manuscript.

10. 198 – *the phrase “linear exciton process” is not quite precise enough. Excitons themselves are not a “process”, they are a quasiparticle. Perhaps “strong in-plane anisotropy of the PL emission” would be better.*

Response 11

We thank the reviewer for the suggestions. We have changed “linear exciton process” to “strong in-plane anisotropy of the PL emission” in the revised manuscript.

James Lloyd-Hughes

We thank Prof. James Lloyd-Hughes for his very valuable reviewing efforts. We hope that our responses have addressed the insightful comments and the revised manuscript is now acceptable for publication.

Reviewer #2 (Remarks to the Author):

In the manuscript "Giant anisotropic photonics in 1D van der Waals semiconductor" LuoJun Du et al. present an study of the photoluminescence, third harmonic generation and Raman spectroscopy of fibers of red phosphorus. The authors find a remarkable anisotropic light emission and third harmonic generation. The manuscript is interesting as more and more part of the community working on 2d materials are getting interested by quasi-1D 2D materials. Fibrous red phosphorus is one example, together with Tellurium or Sb_2Se_3 , of 'real 1D' van der Waals material.

The results are well-presented, the text is clear and the statements made by the authors are sufficiently hold by their results. I missed a bit of more direct comparison with other quasi-1D van der Waals materials (like BP, ReS_2 , $ReSe_2$, TiS_3 , $ZnSe_3$, SnS , $SnSe$, etc...). This would make the manuscript more appealing to the community.

Response 12

We thank the reviewer for the positive assessment of our work and the kind suggestions for improvement. Figure R5 shows the comparison of linear dichroism of fibrous red phosphorus (FRP) with the well-known quasi-1D van der Waals materials (e.g., BP, ReS_2 and $ReSe_2$). The linear dichroism of FRP is large and comparable to the highest values obtained among the well-known quasi-1D van der Waals materials.

Figure R5. The comparison of linear dichroism of fibrous red phosphorus with the well-known quasi-1D van der Waals materials.

To address this comment, Figure R5 has been added into the revised manuscript as Figure 3c. The revised Figure 3 is as follows:

Revised Figure 3. Highly anisotropic PL emission in 1D vdW FRP crystal. **a** PL spectra of an 1D vdW FRP flake under four different polarization configurations with the 532 nm excitation laser. The x (y) axis is parallel (perpendicular) to the b direction of 1D vdW FRP crystal. **b₁-b₇** Polar plots of emission intensity of P2-P8 as a function of polarization angle θ for co-polarized (the polarizations of incident and emission light are parallel to each) and cross-polarized (the polarizations of incident and emission light are perpendicular to each) configurations. The red (blue) solid lines are fitted curves using a $\cos^2 \theta$ ($\sin^2 \theta$) function plus an offset. **c** The comparison of linear dichroism of FRP with the well-known quasi-1D van der Waals materials. The Inset: the linear dichroism of P6 versus the polarization angle θ .

Regarding the thickness of the flakes. Figure 2a is not clear enough to claim that the thinnest flake has 40 nm. It is not clear that the lower-right corner of the image is the substrate and not another part of the flake. Could the authors provide a optical image of that region as well?

Response 13

We thank the reviewer for pointing this out. Figure R6a (also Figure S4 in the supplementary information) is the optical image of a FRP sample corresponding to the AFM image of Figure 2a. To see clearly that the lower-right corner of Figure 2a is the substrate and not another part of the flake, we show the AFM image (Figure R6b) with a larger range than Figure 2a. From Figure R6b, it is clear that the lower-right corner of the AFM image should be the substrate and not another part of the flake.

Figure R6. (a) Optical microscopy image of FRP. (b) AFM image of FRP with a larger view than Figure 2a.

To address this comment, Figure R6b has been added into the revised supplementary information as Figure S4b. The revised Figure S4 is as follows:

Revised Figure S4. (a) Optical microscopy image of FRP. (b) AFM image of FRP with a larger view than Figure 2a in the main text.

It seems that it is difficult to obtain thin ribbons of red phosphorus, the authors should discuss this point to make it clearer to the community. Would liquid phase exfoliation help to get thinner ribbons/fibers?

Response 14

We fully agree with the reviewer. It is difficult to obtain thin ribbons of FRP by mechanical exfoliation. Considering that liquid phase exfoliation with freezing-thawing strategy has successfully produced atomically thin Sb_2Se_3 (also a 1D van der Waals material), it may help to get thinner ribbons/fibers of FRP.

To address this comment, we have added the following sentences in the revised manuscript (page 5):

“It is worth noting that it is difficult to obtain atomically thin FRP by mechanical exfoliation. Considering that liquid phase exfoliation with freezing-thawing strategy has successfully produced ultra-thin Sb_2Se_3 (a 1D vdW material)²⁷, it may help to get atomically thin ribbons/fibers of FRP and deserves further studies.”

Ref. 27: *Adv. Mater.* 29, 1700441 (2017).

In the main text it seems that the material has been synthesized in-house but then in the materials section it is said that has been purchased to 2D Semiconductors. Please revise the text to avoid confusion of the readership.

Response 15

We are sorry for the confusion. The bulk crystals are purchased from 2D Semiconductors. To avoid confusion, we have changed the sentence “Bulk FRP crystals are synthesized through flux zone growth technology (see Methods for more details)” to “Bulk FRP crystals are synthesized through flux zone growth technology (2D Semiconductors, see Methods for more details)” in the revised main text.

In view of this, I believe that the manuscript could be accepted for publication after addressing these minor points.

Response 16

We thank the reviewer again for his/her positive comments on our work. We hope that our responses have addressed the insightful comments raised by the reviewer and the revised manuscript is now acceptable for publication.

Reviewer #3 (Remarks to the Author):

The report highly anisotropic optical properties of fibrous red phosphorus (FRP). This is a 1D vdW material, so there is structural motivation for the proposed study. The presented material needs more careful analysis and not in publishable form. I list my concerns below:

1. The authors claim intense PL emission 40x more than monolayer MoS₂. This statement has no meaning. What is the quantum efficiency of PL emission in FRP? It will be better to make quantitative claims rather than misleading comparisons with monolayer MoS₂.

Response 17

We thank the reviewer for his/her comments. We fully agree with the reviewer that it is better to compare the PL quantum efficiency between the exfoliated FRP sample and monolayer MoS₂. Considering that monolayer MoS₂ possesses an absorbance of ~10% under 2.33 eV excitation [e.g., *Nano Lett.* 13, 3664 (2013)], the PL quantum efficiency of FRP should be more than four times larger than that of monolayer MoS₂, even if we assume that the absorption of FRP is 100%.

To address this comment, we added the following sentence in the revised manuscript (page 6, lines 150-153):

“Considering that monolayer MoS₂ possesses an absorbance of ~10% under 2.33 eV excitation⁴⁴, the PL quantum efficiency of FRP should be more than four times larger than that of monolayer MoS₂, even if we assume that the absorption of FRP is 100%.”

2. What is the exciton life-time in this material? Does the inefficient indirect gap emission still manifest with high efficiency? What is the physical reason behind this effect?

Response 18

We thank the reviewer for his/her comments. Figure R7(a) shows the time-resolved PL spectra measured with a streak camera. Limited by our streak camera setup, we cannot distinguish well multiple PL peaks. The inset of Figure R7(b) is the normalized emission intensity at different wavelengths as a function of time. We can see clearly that the time evolution of normalized emission intensity at different wavelengths coincides with each other, indicating the same life-time for PL emissions at different wavelengths. Figure R7(b) presents that the time evolution of normalized integral intensity from ~600 nm to 700 nm. Through fitting with a single-exponential function [red line of Figure R7(b)], we extract that the PL life-time in FRP is about 10 ps. Note that the PL life-time in FRP (~10 ps) is much longer than the life-time of direct excitation in WSe₂ (~150 fs) [*Nat. Mater.* 14, 889–893 (2015)] and close to the life-time of indirect excitation/phonon replicas in bilayer WSe₂ (~100 ps) [*Nano Lett.* 18, 137-143 (2018)].

Through theoretical calculations, we can know that the valence band along X-Γ-Y direction is quasi-flat, indicating ultra-high density of states (Figures 2e and 2f in the main text). The quasi-flat valence band with large density of states may balance the suppressed transition probability of the indirect transitions and be responsible for the large PL intensity. Further studies (e.g., angle-resolved photoemission spectroscopy) are required to fully understand the origin of the strong PL intensity.

Figure R7. (a) Time-resolved PL spectra measured with a streak camera. (b) The time evolution of normalized integral intensity from 600 nm to 700 nm. Red line is a single-exponential fit. Inset is the time evolution of normalized emission intensity at different wavelengths.

To address this comment, we have added the following sentences in the revised manuscript (page 6, lines 153-161):

“Figure 2c shows the time-resolved PL spectra measured with a streak camera (see Methods for more details). The inset of Figure 2d is the normalized emission intensity as a function of time at different wavelengths. We can see clearly that the time evolution of normalized emission intensity at different wavelengths coincides with each other, indicating the same life-time for PL emissions at different wavelengths. Figure 2d presents that the time evolution of normalized integral intensity from 600 nm to 700 nm. Through fitting with a single-exponential function, we extract that the PL life-time in FRP is about 10 ps. Note that the PL life-time in FRP (~ 10 ps) is much longer than the life-time of direct excitation in WSe_2 (~ 150 fs)⁴⁵ and close to the life-time of indirect excitation/phonon replicas in bilayer WSe_2 (~ 100 ps)⁴⁶.”

Ref. 45: *Nat. Mater.* 14, 889-893 (2015); Ref. 46: *Nano Lett.* 18, 137-143 (2018).

In addition, Figure R7 has been added into the revised manuscript as Figure 2c and Figure 2d. The revised Figure 2 is as follows:

Revised Figure 2. Ultra-strong PL emission intensity in 1D vdW FRP. **a** Upper panel: the typical AFM image of an exfoliated 1D vdW FRP flake, exhibiting several different thicknesses. Lower panel: height profiles taken along the dotted black line in the upper panel. **b** Non-polarized PL spectra (black) for the exfoliated 1D vdW FRP flake in **a**, excited by a 532 nm (2.33-eV) excitation laser at room temperature. The PL spectra of 1D vdW FRP flake are fit to a sum of Lorentzians, from which we can obtain 8 sharp PL peaks, marked as P1-P8. Inset is the energies for the 8 PL peaks. Orange line is the PL spectra of monolayer MoS₂ measured at the same condition. Traces are vertically offset for clarity. **c** Time-resolved PL spectra measured with a streak camera. **d** The time-evolution of normalized integral intensity from 600 nm to 700 nm. Red line is a single-exponential fit. Inset is the time evolution of normalized emission intensity at different wavelengths. **e** The calculated band structure of 1D vdW FRP. Inset shows the BZ path of primitive cell. **f** The mapping of Fermi surface for 1D vdW FRP. Red and cyan isosurfaces represent the highest valence band, and blue and yellow isosurfaces denote the lowest conduction band. The expansion of energy is 40 meV.

3. What is the Debye temperature of this material? Given the complex structure, I assume that the phonons may be populated at room temperature. So, I am even more surprised at the exciton-phonon interaction as the cause for the ordered PL peaks.

Response 19

We thank the reviewer for his/her comments. Debye temperature is an important parameter that characterizes the contribution of acoustic phonons to the specific heat. However, measuring the Debye temperature is currently beyond our scope. For phonon occupation, it is described by the Bose-Einstein statistics: $n(\omega) = \frac{1}{\exp\left(\frac{\hbar\omega}{k_B T}\right) - 1}$, where ω is the frequency of phonon, \hbar denotes the reduced Planck constant, k_B is the Boltzmann constant and

T is the sample temperature. Thus, phonon occupation is determined by the frequency of phonon and temperature, independent on the structure and Debye temperature. For exciton-phonon interaction, it is mainly determined by the exciton-phonon matrix elements, rather than the phonon occupation [e.g., *Rev. Mod. Phys.* 89, 015003 (2017)]. When the exciton-phonon matrix elements are nonvanishing, excitons can couple with phonons, giving rise to the exciton-phonon interaction. In addition, we note that exciton-phonon coupling in PL emission is mainly related to the optical phonons, rather than acoustic phonons. Consequently, phonon replicas should be one possible origin for the multiple PL peaks with equal energy spacing. In addition, we stress that our work focuses on the anisotropy of FRP. The detailed origin of the ordered PL peaks is beyond the scope of our current manuscript and requires further in-depth studies to fully understand.

4. The polarized PL studies show a reduction in intensity, but the broadband emission seems to remain for all conditions. How does this make sense? Can you measure these at low temperatures to see if some of the peaks become sharper. If the exciton-phonon interaction is critical, at low temperatures only the direct emission should be observed.

Response 20

We thank the reviewer for his/her comments. Since the PL emission of FRP is highly anisotropic, the PL intensity detected along y direction would be smaller than that detected along x direction. In addition, because the anisotropy for every PL peak is less than 100%, we can always observe all the PL peaks. Figure R8 shows the temperature dependent PL spectra of FRP. We can observe eight PL peaks at all temperatures. And the PL intensities increase with decreasing the temperature. In addition, it is worth noting that phonon replicas with strong intensity can also occur at low temperature (e.g., 10K). For example, for indirect bilayer WSe₂ (conduction band minimum and valence band maximum are located at Q and K points, respectively), it possesses two transitions: indirect Q-K transition with lower energy and direct K-K transition with higher energy [e.g., *Phys. Rev. Lett.* 124, 217403 (2020) and *Nano Lett.* 18, 137-143 (2018)]. For the indirect Q-K transition, it includes multiple PL peaks (Figure R9a). The origin of these PL peaks is phonon replicas [e.g., *Phys. Rev. Lett.* 124, 217403 (2020), *Nano Lett.* 18, 137-143 (2018) and arXiv:2101.11161]. At low temperature, the phonon replicas in bilayer WSe₂ have stronger intensity than direct K-K transition (Figure R9a). Figure R9b shows the temperature dependent PL spectra of bilayer WSe₂. The intensities of phonon replicas increase with decreasing the temperature. Consequently, phonon replicas with large intensity can be observed at low temperatures.

To address this comment, Figure R8 and Figure R9 are added into the revised supplementary information as Figure S9 and Figure S10, respectively (pages 5 and 6).

Figure R8. Temperature dependent PL spectra of FRP.

Figure R9. (a) PL spectrum of bilayer WSe_2 and its fitting at 10 K. (b) Temperature dependent PL spectra of bilayer WSe_2 .

5. The comparison of third order non-linear optical response to common materials such as gold, GaAs, MoS_2 is misleading. It will be good to compare it with materials that are typically known for this property. It is not necessary that FRP should outperform them, but that is a fair comparison. In these cases, it is beneficial to quantify their performance in terms of a susceptibility coefficient.

Response 21

We thank the reviewer for his/her comments. We fully agree with the reviewer that it is beneficial to quantify the third order non-linear performance in terms of a susceptibility coefficient. In the revised manuscript, we derive the THG third-order susceptibility $\chi^{(3)}$:

$$\chi^{(3)} = \frac{4\varepsilon_0 c^2}{3\omega d} \sqrt{n_\omega^3 n_{3\omega} \frac{I_{3\omega}}{I_\omega^3}} \quad (\text{R2})$$

where ε_0 , c and d are the permittivity of vacuum, speed of light and sample thickness, respectively [e.g., *Phys. Rev. B* 87, 121406(R) (2013)]. And n_ω ($n_{3\omega}$) and I_ω ($I_{3\omega}$) are the refractive index at frequency ω (3ω) and pump (THG) intensity, respectively. Based on Equation (R2), we obtain the THG third-order susceptibility of FRP: $\chi^{(3)}(\text{FRP}) = 2.67 \times 10^{-18} \text{m}^2 \text{V}^{-2}$, which is three times larger than that of MoS_2 ($\chi^{(3)}(\text{MoS}_2) = 8.99 \times 10^{-19} \text{m}^2 \text{V}^{-2}$). Note that calculated refractive indexes n_ω and $n_{3\omega}$ are used to derive the THG third-order susceptibility of FRP because of the absence of experimental results. Figure R10 shows the wavelength dependent refractive index of FRP obtained by density functional

theory calculations. For the pump wavelength of 1300 nm used in Figure 4c of the main text, n_ω and $n_{3\omega}$ are 3.11 and 3.98, respectively.

To address this comment, we have added the following discussion about THG third-order susceptibility $\chi^{(3)}$ in the revised manuscript (page 9):

“In order to better compare the THG between FRP and MoS₂, we derive the THG third-order susceptibility $\chi^{(3)}$:

$$\chi^{(3)} = \frac{4\varepsilon_0 c^2}{3\omega d} \sqrt{n_\omega^3 n_{3\omega} \frac{I_{3\omega}}{I_\omega^3}} \quad (1)$$

where ε_0 , c and d are the permittivity of vacuum, speed of light and sample thickness, respectively⁵⁸. And n_ω ($n_{3\omega}$) and I_ω ($I_{3\omega}$) are the refractive index at frequency ω (3ω) and pump (THG) intensity, respectively. Based on Equation (1), we obtain that the THG third-order susceptibility of FRP is $\chi^{(3)}(\text{FRP}) = 2.672 \times 10^{-18} \text{m}^2 \text{V}^{-2}$, three times larger than that of MoS₂ ($\chi^{(3)}(\text{MoS}_2) = 8.99 \times 10^{-19} \text{m}^2 \text{V}^{-2}$). Note that calculated refractive indexes n_ω and $n_{3\omega}$ [Supplementary Figure S7] are used to derive the THG third-order susceptibility of FRP because of the absence of experimental results.”

Supplementary Figure S7 is Figure R10.

Figure R10. Wavelength dependent refractive index of FRP obtained by theory calculations.

6. *The polarized Raman study is interesting, but it is unclear how that helps us understand the polarized response of the PL study. I see that the symmetry of the Raman response studied for a few peaks may not correspond to the polarized PL response.*

Response 22

We thank the reviewer for his/her comments. Here, we show the polarized Raman results to highlight that the Raman responses of RPF are highly anisotropic as well. In fact, Raman and PL are two completely different processes. Raman scattering is a well-known nonlinear optical process and determined by the derivative of linear susceptibility with respect to the normal coordinate. The polarized Raman responses are determined by the second-rank Raman tensor. While PL is a linear process associated with the recombination of electron-hole pairs. The polarized PL responses are determined by the distribution of the wavefunctions of electrons and holes in real space and have no direct link to the polarized Raman responses.

Overall, I feel that the results presented could be interesting, but the analysis presented is poor. It doesn't help the reader understand the importance of the work. I would not recommend the publication of this article even in a specialized journal.

Response 23

We appreciate the reviewer's positive assessment of our work "the results presented could be interesting" and insightful comments for improvement. We hope that the revised manuscript is now acceptable for publication.

We thank the reviewer for his valuable reviewing efforts.

Yours sincerely

The authors

REVIEWER COMMENTS

Reviewer #1 (Remarks to the Author):

In this revision and response the authors have responded in detail, and satisfactorily, to address the majority of my previous comments. With regard to the current version, I think it is acceptable for publication after the following minor changes are made:

- (1) the title should have a "the" in it, e.g. "Giant anisotropic photonics in **the** 1D van der Waals semiconductor fibrous red phosphorus"
- (2) the abstract needs revision on lines 31-33 in light of the response to my previous comments (and the other reviewer's) that the claim of 40x stronger PL intensity and 100x stronger third harmonic intensity than MoS₂ is not meaningful. Please just state the numbers for the PL quantum efficiency (approximately 4x higher) and the $\chi^{(3)}$ (approximately 3x higher). Those metrics provide a much better comparison than the PL/THG intensity.

James Lloyd-Hughes

Reviewer #2 (Remarks to the Author):

The authors have address the points raised in my referee report. In view of this I recommend the publication of the manuscript.

Reviewer #3 (Remarks to the Author):

The authors have addressed many of my comments and concerns. I am very happy to see that they have taken significant effort to address our concerns. I have listed a few minor issues below. Once these are addressed, I will not have any reservations in recommending the article for publication.

1. I wish to follow up on my earlier comment 5. The response doesn't include the comparison of $\chi^{(3)}$ of materials beyond MoS₂. I am sure there are high non-linearity materials with larger $\chi^{(3)}$, but this comparison can be valuable.
2. Following up on earlier comment, I wonder if the four-wave mixing could play a role in the well order PL peaks. Typically this is a very weak effect, so my guess would be that it is unlikely, but phonon replicas and other origins could also be leading to this effect.
3. The linear dichroism observed in FRP is large but there are other examples with similar or larger materials. I think the LD noted for BaTiS₃ is ~ 1 , seems like an underestimation. Please refer to Wu, J., Cong, X., Niu, S., Liu, F., Zhao, H., Du, Z., Ravichandran, J., Tan, P.-H. & Wang, H. Linear Dichroism Conversion in Quasi-1D Perovskite Chalcogenide. *Adv. Mater.* 31, 1902118 (2019). There is also Sr_{1+x}TiS₃ (Niu, S., Zhao, H., Zhou, Y., Huyan, H., Zhao, B., Wu, J., Cronin, S. B., Wang, H. & Ravichandran, J. Mid-wave and Long-Wave Infrared Linear Dichroism in a Hexagonal Perovskite Chalcogenide. *Chem Mater* 30, 4897–4901 (2018).) with a large dichroic ratio of ~ 20 . I don't think it matters how large or small FRP's linear dichroism value is, but it is good for the community to know the status of various measurements in different materials accurately. It may be worth including in supplementary information a detailed account of wavelength and polarizations considered for these numbers.

Point-by-Point Responses to the Reviewers' Comments

We thank all three reviewers for their positive assessment of our responses and revised manuscript. Reviewer #1 comments *“With regard to the current version, I think it is acceptable for publication after the following minor changes are made”*; Reviewer #2 comments *“The authors have address the points raised in my referee report. In view of this I recommend the publication of the manuscript”* and Reviewer #3 comments *“The authors have addressed many of my comments and concerns.....I will not have any reservations in recommending the article for publication”*.

We also appreciate the reviewers' insightful and constructive comments that help to improve our manuscript. We have addressed all of them in the revised version of our manuscript and in the replies (in blue). We also include a list of changes (in red) made in the main manuscript.

Reviewer # 1 (Remarks to the Author):

In this revision and response the authors have responded in detail, and satisfactorily, to address the majority of my previous comments. With regard to the current version, I think it is acceptable for publication after the following minor changes are made:

Response 1

We thank the reviewer for his positive assessment of our responses and insightful comments to improve our manuscript.

(1) the title should have a "the" in it, e.g. "Giant anisotropic photonics in the 1D van der Waals semiconductor fibrous red phosphorus".

Response 2

We thank the reviewer for the suggestions. We have changed the title to *“Giant anisotropic photonics in the 1D van der Waals semiconductor fibrous red phosphorus”*.

(2) the abstract needs revision on lines 31-33 in light of the response to my previous comments (and the other reviewer's) that the claim of 40x stronger PL intensity and 100x stronger third harmonic intensity than MoS₂ is not meaningful. Please just state the numbers for the PL quantum efficiency (approximately 4x higher) and the chi(3) (approximately 3x higher). Those metrics provide a much better comparison than the PL/THG intensity.

Response 3

We fully agree with the reviewer. We have revised the sentence in the abstract to *“Meanwhile, the photoluminescence (third-harmonic generation) intensity in 1D vdW FRP is strong, with quantum efficiency (third-order susceptibility) four (three) times larger than that in the most well-known 2D vdW materials (e.g., MoS₂)”*.

James Lloyd-Hughes

We thank Prof. James Lloyd-Hughes for his very valuable reviewing efforts. We hope that our responses have addressed the insightful comments and the revised manuscript is now acceptable for publication.

Reviewer #2 (Remarks to the Author):

The authors have address the points raised in my referee report. In view of this I recommend the publication of the manuscript.

Response 4

We thank the reviewer for his/her positive evaluation on our responses and his/her recommendation for publication in *Nature Communications*.

Reviewer #3 (Remarks to the Author):

The authors have addressed many of my comments and concerns. I am very happy to see that they have taken significant effort to address our concerns. I have listed a few minor issues below. Once these are addressed, I will not have any reservations in recommending the article for publication.

Response 5

We thank the reviewer for his/her positive assessment of our responses and insightful comments to improve our manuscript.

1. I wish to follow up on my earlier comment 5. The response doesn't include the comparison of $\chi^{(3)}$ of materials beyond MoS₂. I am sure there are high non-linearity materials with larger $\chi^{(3)}$, but this comparison can be valuable.

Response 6

We thank the reviewer for his/her comments. We fully agree with the reviewer that there must be some materials with third-order susceptibility $\chi^{(3)}$ larger than 1D van der Waals fibrous red phosphorus (FRP). For example, the third-order susceptibility $\chi^{(3)}$ of GaSe is $\chi^{(3)}(\text{GaSe}) = 1.6 \times 10^{-16} \text{m}^2 \text{V}^{-2}$, higher than that of 1D FRP [e.g., *Sci. Rep.* 5, 10334 (2015) and *Adv. Mater.* 30, 1705963 (2018)].

To address this comment, we have added a table in the revised supplementary information (Table S1) to compare the THG third-order susceptibility $\chi^{(3)}$ of FRP with various other typical materials. Table S1 is as follows:

Table S1. The comparison between the THG third-order susceptibility $\chi^{(3)}$ of FRP with various other typical materials.		
Material	$\chi^{(3)}$ ($10^{-19} \text{m}^2/\text{V}^2$)	References
FRP	26.72	This work
MoS ₂	8.99	This work
Graphene	5-10	Ref. 12
Graphene-cavity	400	Ref. 13
Black phosphorus	1.6	Ref. 14
WS ₂	2.4	Ref. 15
MoSe ₂	2.2	Ref. 15
GaSe	1600	Ref. 16
GaTe	2000	Ref. 17
GaAs	14	Ref. 18
TiO ₂	0.21	Ref. 18
Gold	7.6	Ref. 18

LiNbO ₃	0.032	Ref. 19
-------	---------

2. Following up on earlier comment, I wonder if the four-wave mixing could play a role in the well order PL peaks. Typically this is a very weak effect, so my guess would be that it is unlikely, but phonon replicas and other origins could also be leading to this effect.

Response 7

We thank the reviewer for his/her comments. As mentioned by the reviewer, four-wave mixing is a very weak effect and thus is unlikely to produce the well order PL peaks. In fact, in our measurements of third-harmonic generation with much higher peak power than the PL measurements, we did not observe any contribution from four-wave mixing. Therefore, we can indeed rule out the possibility that four-wave mixing is the origin for the well order PL peaks.

3. The linear dichroism observed in FRP is large but there are other examples with similar or larger materials. I think the LD noted for BaTiS₃ is ~1, seems like an underestimation. Please refer to Wu, J., Cong, X., Niu, S., Liu, F., Zhao, H., Du, Z., Ravichandran, J., Tan, P.-H. & Wang, H. *Linear Dichroism Conversion in Quasi-1D Perovskite Chalcogenide*. *Adv. Mater.* 31, 1902118 (2019). There is also Sr_{1+x}TiS₃ (Niu, S., Zhao, H., Zhou, Y., Huyan, H., Zhao, B., Wu, J., Cronin, S. B., Wang, H. & Ravichandran, J. *Mid-wave and Long-Wave Infrared Linear Dichroism in a Hexagonal Perovskite Chalcogenide*. *Chem Mater* 30, 4897-4901 (2018).) with a large dichroic ratio of ~20. I don't think it matters how large or small FRP's linear dichroism value is, but it is good for the community to know the status of various measurements in different materials accurately. It may be worth including in supplementary information a detailed account of wavelength and polarizations considered for these numbers.

Response 8

We thank the reviewer for his/her comments. We fully agree with the reviewer that there are materials with linear dichroism similar to or larger than 1D van der Waals FRP. In Figure 3c of our manuscript (that is, Figure R1 below), we show the comparison of linear dichroism of FRP with the well-known quasi-1D materials (e.g., BP, ReS₂, ReSe₂ and BaTiS₃). It is clear that the linear dichroism of BP (~ 0.93) and BaTiS₃ (~ 0.9) is similar to that of 1D van der Waals FRP (~ 0.91). It is worth noting that the linear dichroism data of BaTiS₃ is taken from Ref. 50, which is exactly the first paper mentioned by the reviewer [e.g., *Linear Dichroism Conversion in Quasi-1D Perovskite Chalcogenide*. *Adv. Mater.* 31, 1902118 (2019)]. Figure R2 [adapted from Ref. 50, *Adv. Mater.* 31, 1902118 (2019)] shows that the linear dichroism of BaTiS₃ (that is, the degree of linear polarization) reaches the maximum value (~ 0.9) at the photon energy of 1.70 eV. Note that linear dichroism is defined as: $\text{linear dichroism} = \frac{I_x - I_y}{I_x + I_y}$, where I_x and I_y denote the photoluminescence emission/absorption/transmission detected in two crystal directions orthogonal to each other. It is clear that linear dichroism should be within the range from -1 to 1.

Considering that some paper utilizes the dichroic ratio to quantify the degree of anisotropy [such as the second paper mentioned by the reviewer, *Chem Mater* 30, 4897-4901 (2018)], rather than the linear dichroism used in our manuscript and Ref. 50 [*Adv. Mater.* 31, 1902118 (2019)], we convert the dichroic ratio into linear dichroism to effectively compare

the anisotropy of different materials (linear dichroism = $\frac{\text{dichroic ratio} - 1}{\text{dichroic ratio} + 1}$). For the second paper mentioned by the reviewer [e.g., Mid-wave and Long-Wave Infrared Linear Dichroism in a Hexagonal Perovskite Chalcogenide. *Chem Mater* 30, 4897-4901 (2018)], the dichroic ratio of ~ 20 corresponds to a linear dichroism of ~0.905.

Figure R1 [Figure 3c of our manuscript]. The comparison of linear dichroism of fibrous red phosphorus with the well-known quasi-1D materials.

Figure R2 [adapted from *Adv. Mater.* 31, 1902118 (2019)]. Absorption coefficients in response to the X- and Y-polarized beam and the degree of linear dichroism in the visible range.

We fully agree with the reviewer that it is worth including a description of the wavelength to which the linear dichroism of different materials corresponds. Note that linear dichroism has already included the information of polarization. To address this, we have added a figure in the revised supplementary information (Figure S14) to include a description

of the wavelength to which the linear dichroism of different materials corresponds. Figure S14 is as follows:

Figure S14. The comparison of linear dichroism of FRP with various other typical materials³⁻¹¹. The horizontal axis represents the energy to which the linear dichroism of different materials corresponds. Note that linear dichroism is defined as: linear dichroism = $\frac{I_x - I_y}{I_x + I_y}$, where I_x and I_y denote the photoluminescence emission/absorption/transmission detected in two crystal directions orthogonal to each other. It is clear that linear dichroism should be within the range from -1 to 1 .

We thank all three reviewers for their valuable reviewing efforts. We believe that the revisions made based upon the reviewers' requests have strengthened our manuscript, and we hope that the revised version now warrants publication in *Nature Communications*.

Yours sincerely
The authors

REVIEWERS' COMMENTS

Reviewer #3 (Remarks to the Author):

The authors have adequately addressed all my concerns. I have no hesitation in recommending the article for publication.

Point-by-Point Responses to the Reviewers' Comments

Reviewer #3 (Remarks to the Author):

The authors have adequately addressed all my concerns. I have no hesitation in recommending the article for publication.

Response

We thank the reviewer for his/her positive evaluation on our responses and his/her recommendation for publication in *Nature Communications*.

Yours sincerely

The authors